# Estimating Hitting Times Locally At Scale

**Themistoklis Haris**
Department of Computer Science
Boston University

**Fabian Spaeh**
Celonis[*]

**Spyros Dragazis**
Department of Computer Science
Boston University

**Charalampos Tsourakakis**
Department of Computer Science
Boston University

## Abstract

Hitting times provide a fundamental measure of distance in random processes, quantifying the expected number of steps for a random walk starting at node $u$ to reach node $v$. They have broad applications across domains such as network centrality analysis, ranking and recommendation systems, and epidemiology.

In this work, we develop local algorithms for estimating hitting times between a pair of vertices $u, v$ without accessing the full graph, overcoming scalability issues of prior global methods. Our first algorithm uses the key insight that hitting time computations can be truncated at the meeting time of two independent random walks from $u$ and $v$. This leads to an efficient estimator analyzed via the Kronecker product graph and Markov Chain Chernoff bounds. We also present an algorithm extending the work of Peng et al. [2021] that introduces a novel adaptation of the spectral cutoff technique to account for the asymmetry of hitting times. This adaptation captures the directionality of the underlying random walk and requires non-trivial modifications to ensure accuracy and efficiency. In addition to the algorithmic upper bounds, we also provide tight asymptotic lower bounds.

We also reveal a connection between hitting time estimation and distribution testing, and validate our algorithms using experiments on both real and synthetic data[1].

## 1 Introduction

Markov chains are a fundamental framework for modeling random processes, with widespread applications across scientific domains including bioinformatics [Krogh, 1998], economics [Chib and Greenberg, 1996], and network science [Xia et al., 2019]. They are particularly prominent in modern machine learning, natural language processing [Almutiri et al., 2022], and the analysis of large-scale social networks [Amati et al., 2018], where the underlying graphs can be massive. In such settings, storing or processing the entire graph globally is often infeasible due to memory or computational constraints. This motivates the development of algorithms that access and process only a small portion of the graph—so-called *local* algorithms.

In this work, we study the problem of computing the *hitting time* statistic between two vertices in an Markov chain. The input is an undirected graph $G$, where we consider simple random walks: from any vertex, the next step is chosen uniformly at random among its neighbors. Given two vertices $u$ and $v$, the hitting time $H_G(u, v)$ is defined as the expected number of steps required for a random walk starting at $u$ to reach $v$ for the first time. This quantity is a fundamental measure in the analysis

---

[1]The code is available at `https://github.com/tkhar/hitting-time-at-scale`.

[2]Work done partially while at Boston University.

39th Conference on Neural Information Processing Systems (NeurIPS 2025).

of random processes and has been extensively studied from various theoretical perspectives [Port and Stone, 1967, Patel et al., 2016, Aldous, 1989, Janson and Peres, 2012, Sauerwald and Zanetti, 2019, Oliveira and Peres, 2019, Feige, 1995, Cohen et al., 2016]. It also serves as a crucial building block in several algorithmic applications, including recommender systems [Cooper et al., 2014] and learning mixtures of Markov chains [Spaeh et al., 2024].

Despite its importance, computing hitting times exactly is computationally expensive, requiring up to $O(n^3)$ time in the worst case [Xia et al., 2019]. To address this, several approaches have been developed that either approximate $H_G(u, v)$ [Cohen et al., 2016] or impose structural assumptions on the input graph to enable more efficient computation [Von Luxburg et al., 2010]. However, these methods are inherently *global*: they require access to the entire graph to produce an estimate. This global nature poses a significant scalability challenge, making such algorithms impractical for massive graphs with millions or billions of nodes.

In this paper, we introduce a suite of algorithms that estimate hitting times *locally*, by executing a small number of short random walks centered around the vertices $u$ and $v$. We rigorously analyze these algorithms and formally prove that they achieve approximation guarantees comparable to those of global methods—while operating with significantly lower time complexity and without requiring access to the full graph.

**Related work**   Our work is closely related to that of Peng et al. [2021], who design local algorithms for computing the *effective resistance* $R_{\text{eff}}(u, v)$ – a distance measure that captures the voltage difference between $u$ and $v$ when one unit of current is injected at $u$ and edges act as electrical resistors. Their approach is based on spectrally decomposing the random walk transition matrix, interpreting $R_{\text{eff}}(u, v)$ as a light-tailed sum that can be efficiently truncated. Effective resistance is linked to the hitting time as $H_G(u, v) + H_G(v, u) = 2mR_{\text{eff}}(u, v)$. As a result, the hitting time can be efficiently estimated using the effective resistance in graphs where $H_G(u, v) \approx H_G(v, u)$, such as vertex transitive graphs. However, real-world networks exhibit an asymmetric and skewed hitting time distribution, meaning that new ideas beyond the work of Peng et al. [2021] need to be introduced.

We also build directly on the work of Cohen et al. [2016], who derive an identity expressing the hitting time in terms of the Laplacian of the graph $G$. While powerful, their algorithm remains global in nature. Finally, a key component of our method is the connection between hitting time and the notion of *meeting time*—the expected time for two independent random walks, starting at $u$ and $v$, to meet. Related concepts such as the *coalescence* time have been studied extensively and are known for many specific graph families [Kanade et al., 2023, Cooper et al., 2013, Oliveira, 2012]. We utilize this connection from an algorithmic lens, using it to obtain an efficient and practical estimator.

**Our Results**   We summarize our main contributions below:

1. We design and analyze an efficient local algorithm for estimating hitting times, based on a novel analysis of meeting times (Algorithm 1). We demonstrate its practical effectiveness through extensive experiments on both synthetic and real-world datasets (Section 6).

2. We extend the spectral cutoff technique introduced by Peng et al. [2021] to derive an alternative local algorithm for hitting time estimation (Algorithm 3). While generally less efficient than our meeting-based method, it can outperform it in certain settings.

3. We provide a detailed theoretical study of the trade-off between approximation and sample complexity in hitting time estimation, including both upper and lower bounds (Section 5).

4. We uncover a theoretical connection between hitting time estimation and sublinear-time property testing (Subsection 4.1).

## 2   Preliminaries

Let $G = (V, E)$ be an undirected graph with $(n, m) = (|V|, |E|)$ and $A \in \{0, 1\}^{n \times n}$ be its adjacency matrix. Let $D$ be a diagonal matrix containing the degrees of the vertices in $V$ in its diagonal. Let $P = D^{-1}A$ be the row-stochastic transition matrix of $G$. Throughout, we assume that $G$ is connected and the associated random walk aperiodic. It is known that the transition matrix has a full spectrum

$\lambda_1 = 1 > \lambda_2 \geq \cdots \geq \lambda_n > -1$. A random walk in $G$ is a sequence of vertices $(X_t)_{t \geq 0}$ such $X_t \in V$ and

$$\Pr[X_{t+1} = v \mid X_t = u] = \frac{1}{\deg(u)} = P_{uv}$$

We also let $\pi$ be the stationary distribution of $G$, where $\pi_v = \frac{\deg v}{m}$. It can be shown that any initial distribution over the vertices eventually converges to $\pi$:

**Definition 2.1** (Mixing Time). *The $\varepsilon$-**mixing time** of $G$ is defined as:*

$$t_{\text{mix}} = t_{\text{mix}}(\varepsilon) = \min\{t : \max_{x \in \Delta_n} ||xP^t - \pi||_{TV} \leq \varepsilon\}$$

**Definition 2.2** (Hitting Time). *Let $u, v \in V$. The **hitting time** $H_G(u,v)$ is the expected number of steps required to reach $v$ from $u$. In other words, if we let $T = \min\{t \geq 0 : X_t = v\}$ then*

$$H_G(u,v) = \mathbb{E}[T \mid X_0 = u].$$

The following useful Lemma expresses $H_G(u,v)$ in terms of the Laplacian $I - P^\top$. Let us use $\mathbf{e}_v \in \mathbb{R}^n$ to denote the $v$-th vector of the standard basis.

**Lemma 2.1** ([Cohen et al., 2016]). *It is true that:*

$$H_G(u,v) = \left(\mathbf{1} - \frac{1}{s_v}\mathbf{e}_v\right)^\top (I - P^\top)^+ \chi_{uv} \tag{1}$$

We define the **effective resistance** $R_{\text{eff}}(u,v)$ in terms of the hitting time, although other equivalent definitions also exist [Spielman, 2019]:

$$R_{\text{eff}}(u,v) = \frac{1}{2m}(H_G(u,v) + H_G(v,u)) \tag{2}$$

We shall make scarce use of the *Kronecker Product* between two graphs.

**Definition 2.3** (Kronecker Product). *The **Kronecker product** of two graphs $G = (V_G, E_G)$ and $H = (V_H, E_H)$ is defined as a graph $G \times H = (V_G \times V_H, E_{G \times H})$ where $(\{u,v\}, \{w,z\}) \in E_{G \times H}$ if and only if $(u,w) \in E_G$ and $(v,z) \in E_H$.*

Additional preliminary definitions and results are shown in Appendix A.

## 3 A Meeting Time Perspective

Our primary contribution is an algorithm that estimates the hitting time by relating it to the *meeting time* of two parallel random walks starting at $u$ and $v$. We simulate the walks step-by-step, accumulating terms in the hitting time sum until they meet. Due to the cancellation structure of the infinite sum defining $H_G(u,v)$, this truncated computation suffices. Our algorithm is given as pseudocode in Algorithm 1.

To analyze Algorithm 1, we first bound the meeting time in probability using the mixing time:

**Lemma 3.1.** *Let $(X_t)_{t \geq 0}$ and $(Y_t)_{t \geq 0}$ be two random walks starting at states $X_0 = u$ and $Y_0 = v$. Let $T = \min\{t \geq 0 : X_t = Y_t\}$ be the meeting time of the two random walks. Then,*

$$\Pr[T > t] \leq O\left(\frac{1}{\sqrt{\pi_G(u)\pi_G(v)}} \exp\left(-\frac{||\pi_G||_2^2 t}{72 t_{\text{mix}}}\right)\right)$$

*Proof.* The proof uses a Markov Chain concentration bound on the Kronecker product $G \times G$ and is included in Appendix B. □

Next, we argue that if we make $t_{\text{max}}$ large enough, all the random walks will, with high probability, meet before then.

**Lemma 3.2.** *Algorithm 1 outputs* Failure *with probability at most $\frac{1}{n}$.*

---

**Algorithm 1:** Estimating the Hitting Time via Meeting Times

---

1 **Input:** Graph $G = (V, E)$, vertices $u, v \in V$, mixing time $t_{\mathrm{mix}}$, accuracy $\varepsilon$
2 **Output:** Estimate $\widetilde{H}_{uv}$ of the hitting time $H_G(u, v)$
3 Define

$$t_{\max} = \frac{100 \cdot t_{\mathrm{mix}} \ln\left(\frac{n}{\pi_G(u)\pi_G(v)}\right)}{\|\pi_G\|_2^2} \quad \text{and} \quad \ell = \frac{2t_{\max}^2 \ln n}{\pi_G^2(v)\varepsilon^2}$$

4 Initialize random walks $X_0^{(i)} = u$ for $i \in I$ with $I = \{1, 2, \ldots, \ell\}$
5 Initialize random walks $Y_0^{(j)} = v$ for $j \in J$ with $J = \{1, 2, \ldots, \ell\}$
6 Let $\widetilde{H}_{uv} \leftarrow 0$
7 **for** $t = 0, 1, 2, \ldots, t_{\max}$ **do**
8      Let $x_w := |\{i \in I : X_t^{(i)} = w\}|$ and $y_w := |\{j \in J : Y_t^{(j)} = w\}|$ for each $w \in V$.
9      **for** $w \in V$ **do**
10          $z = \min\{x_w, y_w\}$
11          Remove $z$ arbitrary indices $i \in I$ for which $X_t^{(i)} = w$ and $z$ indices $j \in J$ for which
         $Y^{(j)} = w$.
12      Update

$$\widetilde{H}_{uv} \leftarrow \widetilde{H}_{uv} + \frac{y_v - x_v}{\ell \cdot \pi_G(v)}$$

13      Advance the random walks $X^{(i)}$ for $i \in I$ and $Y^{(j)}$ for $j \in J$ by one step as defined by $G$.
14 **if** $I = \emptyset$ **then**
15      **return** $\widetilde{H}_{uv}$
16 **else**
17      **return** Failure

---

*Proof.* Let us pretend that random walks in Algorithm 1 continue after meeting and are not eliminated. Let then $T_{i,j} = \min\{t \geq 0 : X_t^{(i)} = Y_t^{(j)}\} \in \mathbb{N}_{\geq 0}$ be the meeting time of the of the $i$-th random walk starting from $u$ and the $j$-th random walk starting from $v$.

Let $\mathcal{E}$ be the event corresponding to $T_{i,j} \leq t_{\max}$ for all $i, j \in [\ell]^2$. We claim that, conditioned on $\mathcal{E}$, Algorithm 2 does not output Failure. Consider a bipartite graph $G_B = (V_B, E_B)$ where $V_B = A \times B$ with $|A| = |B| = \ell$ correspond to the random walks $X$ and $Y$. The edges of this graph represent the meeting event of two random walks and they are provided in an online manner: in timestep $t = 0, 1, ..., t_{\max}$ we see observe a set of random walks in $A$ meeting another set of random walks in $B$ for the first time. In other words, we observe all the edges $A_t \times B_t$ where $A_t \subseteq A$ and $B_t \subseteq B$. Once an edge has been given, it is not given again, so this is a partition of the edge set of the graph.

Since $\mathcal{E}$ occurs, the bipartite graph is complete and $\{A_t \times B_t\}_{t=0}^{t_{\max}}$ is a partition of all the edges. Our algorithm tries to produce a matching $\sigma$ between $A$ and $B$ by greedily matching and removing vertices as the edges appear in batches. We output Failure if and only if we fail to output a matching. We claim that, given $\mathcal{E}$, we will always find a matching.

This can be seen via an inductive argument: we claim that before any timestep $t$ we have a valid matching over all vertices that have appeared as part of previously revealed edges. This easily holds at $t = 0$ because the initial matching is empty. Suppose then that it holds for some $t$. When the edges $A_t \times B_t$ are revealed, consider the subsets $A_t' \subseteq A_t$ and $B_t' \subseteq B_t$ of unmatched nodes. There has to exist a perfect matching in $A_t' \times B_t'$ because all those edges are revealed. We pick any such matching arbitrarily and so our matching is successfully extended.

Finally, we need to bound the probability that $\mathcal{E}$ holds. We know that for any $i, j \in [\ell]$ we have:

$$\Pr[T_{i,j} > t_{\max}] \leq O\left(\sqrt{\pi(u)\pi(v)} \exp\left(-\frac{\mu t_{\max}}{72 t_{\mathrm{mix}}}\right)\right).$$

Hence, by a union bound over $\binom{\ell}{2}$ pairs of random walks:

$$\Pr[\neg\mathcal{E}] \leq O\left(\frac{\ell^2}{\sqrt{\pi_G(u)\pi_G(v)}} \exp\left(-\frac{\|\pi_G\|_2^2 t_{\max}}{72 t_{\mathrm{mix}}}\right)\right) = \frac{1}{n}$$

by our choice of $t_{\max}$ and $\ell$, where we can assume $\ell \leq n^3$ as an upper bound. $\qquad\square$

Next, we express the hitting time as a function of two random walks that stop upon meeting. To that end, we first show a structural result similar to [Peng et al., 2021]: By spectrally decomposing the Laplacian matrix, we can express the hitting time as an infinite series, from which we can subsequently truncate the tail.

**Lemma 3.3.** *Let $W := P^\top = AD^{-1}$. The following identity holds:*

$$H_G(u, v) = \sum_{i=0}^{\infty} \left(\mathbf{1} - \frac{1}{\pi(v)}\mathbf{1}_v\right)^\top W^i \chi_{uv} \tag{3}$$

We prove this lemma shortly, in Section 4. With it in place, our meeting time analysis can proceed.

**Lemma 3.4.** *Let $(X_t)_{t \geq 0}$ and $(Y_t)_{t \geq 0}$ be two random walks in $G$ starting from $X_0 = u$ and $Y_0 = v$, respectively. Let $T = \min\{t \geq 0 : X_t = Y_t\}$ be a random variable for the meeting time of the two random walks. The following holds:*

$$H_G(u, v) = \mathbb{E}\left[\sum_{t < T} \left(\mathbb{1}_{[Y_t=v]} - \mathbb{1}_{[X_t=v]}\right)\right].$$

*Proof.* The hitting time formula of Lemma 2.1 can be written as an infinite sum. Algebraic manipulation then gives us that:

$$H_G(u, v) = \sum_{t=0}^{\infty} \left(\mathbf{1} - \frac{1}{\pi_G(v)}\mathbf{1}_v\right)^\top W^t \chi_{uv}$$

$$= \sum_{t=0}^{\infty} \left(\underbrace{\sum_{w \in V} \Pr[u \to^t w]}_{=1} - \frac{1}{\pi_G(v)}\Pr[u \to^t v] - \underbrace{\sum_{w \in V} \Pr[v \to^t w]}_{=1} + \frac{1}{\pi_G(v)}\Pr[v \to^t v]\right)$$

$$= \frac{1}{\pi_G(v)}\sum_{t=0}^{\infty} \left(\Pr[v \to^t v] - \Pr[u \to^t v]\right) \qquad \text{(because $G$ is connected)}$$

Due to memorylessness, $(X_t)_{t \geq T}$ and $(Y_t)_{t \geq T}$ have the same distribution. Hence,

$$\pi_G(u)H_{uv} = \sum_{t=0}^{\infty} \left(\Pr[v \to^t v] - \Pr[u \to^t v]\right)$$

$$= \mathbb{E}\left[\sum_{t=0}^{\infty} \left(\mathbb{1}_{[Y_t=v]} - \mathbb{1}_{[X_t=v]}\right)\right]$$

$$= \mathbb{E}\left[\sum_{t < T} \left(\mathbb{1}_{[Y_t=v]} - \mathbb{1}_{[X_t=v]}\right)\right] + \underbrace{\mathbb{E}\left[\sum_{t \geq T} \left(\mathbb{1}_{[Y_t=v]} - \mathbb{1}_{[X_t=v]}\right)\right]}_{=0}$$

$$= \mathbb{E}\left[\sum_{t < T} \left(\mathbb{1}_{[Y_t=v]} - \mathbb{1}_{[X_t=v]}\right)\right]$$

$\qquad\square$

We conclude our analysis by showing that our algorithm approximates $H_G(u, v)$ well.

**Lemma 3.5.** *If Algorithm 1 does not fail, it outputs an estimate $\widetilde{H} := \widetilde{H}_G(u,v)$ where $|\widetilde{H}_G(u,v) - H_G(u,v)| \le \varepsilon$ with probability at least $1 - 1/n$.*

*Proof.* Let us denote with $(X_t^{(i)})_{t \ge 0}$ and $(Y_t^{(i)})_{t \ge 0}$ the $i$-th random walk starting form $u$ and $v$, respectively. Let $A_i = |\{0 \le t < T : X_t^{(i)} = v\}|$ be the number of times that the $i$-th random walk $X^{(i)}$ visits $v$ before it gets eliminated by meeting at time $T$. We define $B_i$ accordingly for $Y^{(i)}$, which allows us to write the output of Algorithm 1 as

$$\widetilde{H} = \frac{1}{\ell} \sum_{i=1}^{\ell} \frac{1}{\pi_G(v)} (B_i - A_i)$$

and we immediately see that $\mathbb{E}[\widetilde{H}] = H_G(u,v)$ by Lemma 3.4. If the algorithm does not report failure, we know that $A_i, B_i \le t_{\max}$ and hence $|B_i - A_i| \le 2t_{\max}$. By a Hoeffding bound,

$$\Pr\left[|\widetilde{H} - H_G(u,v)| \ge \varepsilon\right] \le 2\exp\left(-\frac{\pi_G^2(v)\ell\varepsilon^2}{2t_{\max}^2}\right) = \frac{1}{n}$$

for our choice of $\ell$. $\qquad\square$

Finally, we arrive at the following Theorem:

**Theorem 3.1.** *There exists an algorithm that, given any $u, v \in V$ outputs, with probability at least $1 - 2/n$, an estimate $\widetilde{H}$ such that $|\widetilde{H} - H_G(u,v)| \le \varepsilon$ and has total runtime*

$$O\left(\frac{t_{\mathrm{mix}}^3}{\|\pi_G\|_2^6 \cdot \pi_G^2(v) \cdot \varepsilon^2} \log^3\left(\frac{n}{\pi_G(u) \cdot \pi_G(v)}\right)\right)$$

*Proof.* The runtime follows directly from Lemmata 3.2 and 3.5 with a union bound. $\qquad\square$

## 3.1 Effective Resistance Calculation

Having established Theorem 3.1, we can use it to calculate the effective resistance $R_{\mathrm{eff}}(u,v)$.

---
**Algorithm 2:** Estimating the Effective Resistance via Meeting Times

---
1 **Input:** Graph $G = (V, E)$, vertices $u, v \in V$, mixing time $t_{\mathrm{mix}}$, accuracy $\varepsilon$
2 **Output:** Estimate $\widetilde{R}$ of the effective resistance $R_G(u,v)$
3 $\tilde{H}_{uv} \leftarrow \text{HITTINGTIME}(G, u, v, t_{\mathrm{mix}}, \varepsilon m/2)$
4 $\tilde{H}_{vu} \leftarrow \text{HITTINGTIME}(G, v, u, t_{\mathrm{mix}}, \varepsilon m/2)$
5 $\tilde{R} \leftarrow \frac{1}{2m}(\tilde{H}_{uv} + \tilde{H}_{vu})$
6 **return** $\tilde{R}$

---

**Corollary 3.1.** *There exists an algorithm that, given any $u, v \in V$ outputs, with probability at least $1 - 1/n$ an estimate $\widehat{R}$ such that $|\widehat{R} - R_{\mathit{eff}}(u,v)| \le \varepsilon$ and has runtime*

$$O\left(\frac{t_{\mathrm{mix}}^3}{\|\pi_G\|_2^6 \cdot \mu_{u,v}^2 \cdot \varepsilon^2 m^2} \log^3\left(\frac{n}{\pi_G(u) \cdot \pi_G(v)}\right)\right)$$

*where $\mu_{u,v} = \min\{\pi_G(u), \pi_G(v)\}$.*

*Proof.* Running Algorithm 1 with $\varepsilon' = \frac{\varepsilon}{2} \cdot 2m$ we get that:

$$|\widehat{R} - R_{\mathrm{eff}}(u,v)| = \frac{1}{2m}|(\widetilde{H}_{uv} - H_G(u,v)) + (\widetilde{H}_{vu} - H_G(v,u))| \le \varepsilon$$

The runtime follows from the runtime of Algorithm 1 $\qquad\square$

# 4 Hitting Times via Spectral Cutoff

In this section we describe yet another local algorithm for estimating the hitting time $H_G(u, v)$ between two vertices $u$ and $v$. We first prove the spectral decomposition result from earlier.

**Lemma 4.1.** *Let $W := P^\top = AD^{-1}$. The following identity holds:*

$$H_G(u, v) = \sum_{i=0}^{\infty} \left( \mathbf{1} - \frac{1}{\pi(v)} \mathbf{1}_v \right)^\top W^i \chi_{uv} \tag{4}$$

*Proof.* Let $\lambda_1 \geq \lambda_2 \geq \cdots \geq \lambda_n$ be the spectrum of $W$, so the eigendecomposition of $W$ is:

$$W = \sum_{j=1}^{n} \lambda_j w_j w_j^\top$$

where the vectors $w_1, ..., w_n$ form an orthonormal basis. Note that $\lambda_1 = 1$, so we can write the pseudo-inverse of $I - W$ as:

$$(I - W)^+ = \sum_{j=2}^{n} \frac{1}{1 - \lambda_j} w_j w_j^\top$$

Then we have by the sum of an infinite geometric series that:

$$(I - W)^+ = \sum_{j=2}^{n} \sum_{s=0}^{\infty} \lambda_j^s w_j w_j^\top = \sum_{s=0}^{\infty} \sum_{j=2}^{n} \lambda_j^s w_j w_j^\top = \sum_{s=0}^{\infty} (W^s - w_1 w_1^\top)$$

Now we substitute back to the hitting time calculation of Lemma 2.1:

$$H_G(u, v) = \sum_{j=0}^{\infty} \left( \mathbf{1} - \frac{1}{\pi(v)} \mathbf{e}_v \right)^\top (W^j - w_1 w_1^\top) \chi_{uv} = \sum_{j=0}^{\infty} \left( \mathbf{1} - \frac{1}{\pi(v)} \mathbf{e}_v \right)^\top W^j \chi_{uv} \tag{5}$$

since $w_1^\top \chi_{uv} = 0$ because $w_1 = \frac{1}{\sqrt{n}} \mathbf{1}_n$. $\qquad\square$

Our local algorithm for estimating the hitting time $H_B(u, v)$ relies on Lemma 4.1. We identify a threshold $\ell$ such that truncating the sum $\sum_{j=\ell}^{\infty} (\mathbf{1} - \frac{1}{\pi(v)} \mathbf{e}_v)^\top W^j \chi_{uv}$ incurs additive error up to $\frac{\varepsilon}{2}$ from the total sum. Then, we estimate the rest using a collection of independent random walks.

---

**Algorithm 3:** A "Cutoff" Algorithm for Estimating Hitting Times

---

1 **Inputs:** Adjacency matrix and degree query access to graph $G$, Vertices $u, v \in V$.

2 **Output:** Estimate $\widehat{h}_G(u, v)$ of the hitting time $H_G(u, v)$.

3 **Parameters:** Spectral gap $\lambda \in [-1, 1), \varepsilon > 0$

4 Let $\ell \leftarrow \frac{\log \frac{n}{\varepsilon - \varepsilon\lambda}}{\log(1/\lambda)}$.

5 Initialize $\widehat{h}_G(s, t) \leftarrow 0$.

6 **for** $i = 0$ *to* $\ell - 1$ **do**

7 $\quad$ Let $r \leftarrow \frac{32\ell^2 \log(40\ell)}{\varepsilon^2 \pi(v)^2}$

8 $\quad$ Execute $r$ random walks from $v$ of length $i$ and let $T_u, T_v$ be the number of those that end in $\quad$ $u$ and $v$ respectively.

9 $\quad$ Let $\widehat{p_{i,u}} \leftarrow \frac{2m}{\deg(v)} \cdot \frac{T_u}{r}$ and $\widehat{p_{i,v}} \leftarrow \frac{2m}{\deg(v)} \cdot \frac{T_v}{r}$

10 $\quad$ Update $\widehat{h}_G(u, v) \leftarrow \widehat{h}_G(u, v) + (\widehat{p_{i,v}} - \widehat{p_{i,u}})$

11 **Output** $\widehat{h}_G(u, v)$

---

**Theorem 4.1.** *Algorithm 3 returns an estimate $\widehat{h}_G(s,t)$ such that with probability at least $9/10$ it holds that:*

$$\left|\widehat{h}_G(s,t) - H_G(s,t)\right| \le \varepsilon$$

*Its runtime is bounded by*

$$O\left(\frac{\ell^4}{\varepsilon^2 \cdot \pi_G(v)^2} \log \ell\right), \quad where \quad \ell = \frac{\log \frac{n}{\varepsilon - \varepsilon\lambda}}{\log \frac{1}{\lambda}}$$

*for the spectral gap $\lambda := \max\{|\lambda_2|, |\lambda_n|\}$ of $G$.*

*Proof.* The proof is given in Appendix C. $\square$

**Remark 4.1.** *Using this algorithm in the formula $R_{\text{eff}}(u,v) = \frac{1}{2m}(H_G(u,v) + H_G(v,u))$ gives us an algorithm with roughly the same runtime is Algorithm 1 of Peng et al. [2021].*

### 4.1 A Connection with Property Testing

In Algorithm 3, we define the "cutoff" point for the hitting time sum by spectrally decomposing the Laplacian matrix, as done by Peng et al. [2021]. However, this approach requires knowing the spectral gap $\lambda$, which can only be computed by analyzing the entire graph, and is related to the Markov Chain's mixing time, which may be arbitrarily large.

To address this, we propose a local truncation method. Intuitively, if the states $u$ and $v$ are "close", random walks from both should converge quickly, even with a large mixing time. We introduce a local mixing time $t_{\min}^{(\varepsilon)}(u,v)$ and show that when the Markov Chain converges quickly, the cutoff point is determined by this local mixing time. This result follows from a convergence lemma for ergodic Markov Chains [Freedman, 2017]. Though we focus on the effective resistance problem for simplicity, the theory extends to hitting times as well.

The challenge then becomes determining $t_{\min}^{(\varepsilon)}(u,v)$, which we solve using property testing algorithms [Batu et al., 2013, Chan et al., 2014] to perform a binary search for a good upper bound. These algorithms run in sublinear time, making our method also sublinear. Under mild convergence assumptions, this approach removes the need for $\lambda$ and $t_{\min}$, yielding a fully local algorithm. Details of this theoretical development can be found in Appendix F.

## 5 The Optimality of Walk Sampling

In Appendix D we use a Chernoff bound for Markov Chains to analyze possibly the simplest local algorithm for hitting time estimation: sample a number of arbitrarily long random walks and return the average hitting time. Our analysis yields the following result:

**Theorem 5.1.** *Let $M$ be an ergodic Markov chain with stationary distribution $\pi$. Let $s$ and $t$ be two different states. There exists an algorithm calculating the hitting time $H_G(s,t)$ to within an absolute error of $\varepsilon$ with probability at least $\frac{1}{n}$ that uses*

$$\widetilde{O}\left(\frac{t_{\text{mix}}^2}{\pi(t)^2 \varepsilon^2}\right)$$

*random walk samples of length at most $\widetilde{O}(t_{\text{mix}}/\pi(t))$.*

Could we have sampled fewer random walks in a similar fashion and aggregated them to obtain a good estimate? In Appendix E we answer this question in the negative: the walk sampling algorithm has optimal sample complexity. Our proof involves a carefully constructed "barbell"-like graph and utilizes anti-concentration arguments for the geometric and binomial distributions.

**Theorem 5.2.** *Suppose an algorithm is able to estimate $H_G(u,v)$ within additive error $\varepsilon$ with constant probability of success by taking the average over $r$ random walk samples of unbounded length. Then, in the worst case, we must have that:*

$$r = \Omega\left(\frac{t_{\text{mix}}^2}{\pi(v)^2 \varepsilon^2}\right)$$

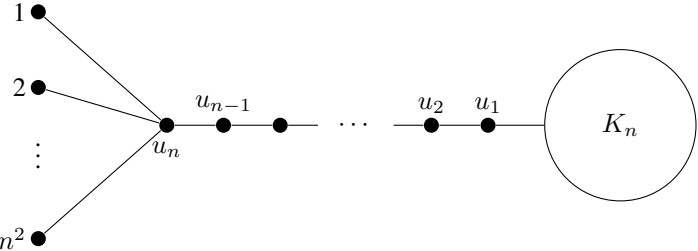

Figure 1: A 'barbell''-like graph: a star, a path and a clique

These results demonstrate that estimating hitting times is inherently computationally hard. Nevertheless, our local algorithms—despite having comparable theoretical runtime to the sampling approach—consistently outperform it in practice.

# 6 Experimental Results

In this section, we perform a comparative study of different algorithms for estimating the hitting time $H_G(u, v)$ between vertices $u$ and $v$. Our experiments are conducted in Python 3.6 with Numba 0.53 on a 2.9 GHz Intel Xeon Gold 6226R processor with 384GB of RAM.

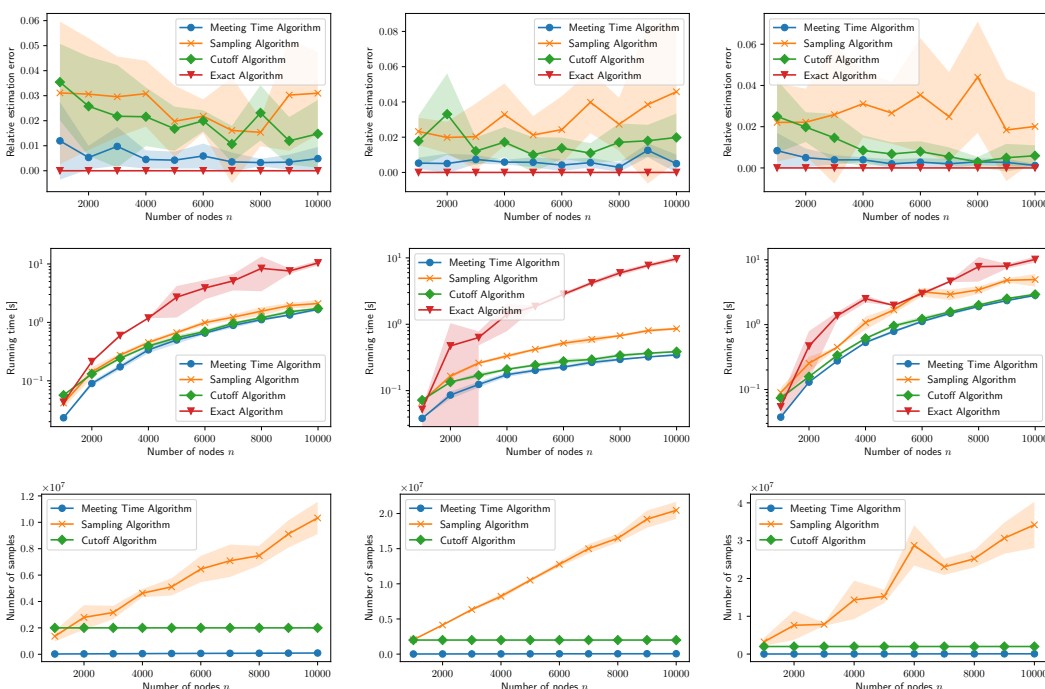

Figure 2: Hitting time estimation on synthetic networks as a function of the number of nodes. Depicted are: Barabasi-Albert graphs (*left*), Erdos-Renyi graphs (*middle*) and Stochastic Block Model (SBM) Graphs (*right*).

We compare between Algorithm 1, Algorithm 3, the walk sampling algorithm and an exact solver which determines $H_G(u, v)$ by solving a linear system: Let $H \in \mathbb{R}^n$ be such that $H_w = H_G(w, v)$. Then, for every $u \in V$ we have: $H_u = H_G(u, v) = \sum_{w \in N(v)} \frac{1}{\deg(v)} H_w$.

Additional information on our experimental setting, experiments with parallelization, and other ablation studies can be found in Appendix G.

**Synthetic Datasets**  We first compare the performance and error of each of those algorithms on synthetic graph datasets. We focus on three random graph models: *Erdős-Rényi*, where each pair of vertices forms an edge with probability $p$, *Barabási-Albert*, where a preferential attachment mechanism is used, and the *Stochastic Block Model* (SBM), where we model the emergence of community structures. We estimate the hitting time from the first to the last node. We compare the relative error, running time and number of sample walks.

Our experiments show that our algorithms estimate the hitting time with low relative error. Among the methods evaluated, the naive walk-sampling algorithm performs the worst, requiring more samples and yielding weaker error guarantees. In contrast, the meeting-time-based algorithm is highly efficient while also maintaining very low error. It also notably exhibits much lower variance in the error than both the sampling algorithm and the cutoff algorithm.

**Real World Large Graphs**  We also run our algorithms on real world graphs, specifically the American Football Division IA games graph [Girvan and Newman, 2002] and the SNAP Facebook and Twitter graphs [Leskovec and Mcauley, 2012]. To stress test our algorithms adequately, we randomly sample pairs $u, v$ uniformly, but also proportionally and inversely proportional according to the product of degrees or Pagerank centralities Page et al. [1999]: Figure 5 in Appendix G shows the correlation between hitting times and the degree and Pagerank centrality products. We find that Algorithm 1 consistently achieves low error with small variance. Note also that we were not able to compute the true hitting time for the Twitter via a (sparse) linear system. Thus, we report the mean absolute and mean squared error compared to 100 iterations of the meeting time algorithm, as it is the only algorithm which is unbiased.

Table 1: Relative estimation error for different pair sampling strategies.

| | Algorithm | $\deg(u) \cdot \deg(v)$ prop. | inv-prop. | $\mathrm{pagerank}(u) \cdot \mathrm{pagerank}(v)$ prop | inv-prop. | uniform |
|---|---|---|---|---|---|---|
| Football | **Meeting Time Alg.** | $0.013 \pm 0.009$ | $0.012 \pm 0.009$ | $0.012 \pm 0.009$ | $0.011 \pm 0.009$ | $0.012 \pm 0.009$ |
| | Cutoff Alg. | $1.061 \pm 0.778$ | $1.047 \pm 0.805$ | $1.061 \pm 0.806$ | $0.983 \pm 0.763$ | $0.994 \pm 0.778$ |
| | Sampling Alg. | $0.025 \pm 0.019$ | $0.025 \pm 0.019$ | $0.026 \pm 0.019$ | $0.025 \pm 0.019$ | $0.025 \pm 0.019$ |
| Facebook | **Meeting Time Alg.** | $0.011 \pm 0.010$ | $0.011 \pm 0.010$ | $0.016 \pm 0.014$ | $0.016 \pm 0.017$ | $0.012 \pm 0.011$ |
| | Cutoff Alg. | $0.078 \pm 0.096$ | $0.108 \pm 0.129$ | $0.253 \pm 0.297$ | $0.193 \pm 0.357$ | $0.147 \pm 0.156$ |
| | Sampling Alg. | $0.026 \pm 0.020$ | $0.025 \pm 0.021$ | $0.030 \pm 0.025$ | $0.026 \pm 0.021$ | $0.027 \pm 0.021$ |
| Twitter | **Meeting Time Alg.** | $0.002 \pm 0.001$ | $0.002 \pm 0.002$ | $0.010 \pm 0.007$ | $0.009 \pm 0.007$ | $0.005 \pm 0.004$ |
| | Cutoff Alg. | $0.022 \pm 0.016$ | $0.023 \pm 0.017$ | $0.162 \pm 0.126$ | $0.102 \pm 0.077$ | $0.086 \pm 0.066$ |
| | Sampling Alg. | $0.025 \pm 0.019$ | $0.024 \pm 0.018$ | $0.026 \pm 0.019$ | $0.027 \pm 0.020$ | $0.024 \pm 0.018$ |

## Limitations

Our local algorithms approximate $H_G(u, v)$, with accuracy depending on the graph's structure—a typical trade-off for efficiency. Our algorithms can be thus less efficient when the mixing time is large, though this also affects prior methods.

## 7   Conclusion

We presented theoretical and empirical results on efficient estimators for hitting times in random walks, bridging Markov chain theory with sublinear-time algorithms. Future research directions include establishing universal lower bounds, and exploring applications of such algorithms in domains like recommender systems.

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

# A    Additional Preliminaries

In this Section we deposit additional preliminary definitions and lemmata essential to our main exposition.

**Definition A.1** (Ergodic Markov Chains). *A Markov Chain is **ergodic** if it is aperiodic and positive recurrent. It is aperiodic when the gcd of all possible return times to any state is equal to 1, and it is positive recurrent if the expected return time to any state is finite.*

We shall assume that the Markov chain defined on $G$ is ergodic, and that $G$ is connected with minimum vertex degree at least 1. The following theorem describes the rate of convergence of an ergodic Markov Chain to its stationary distribution $\pi$.

**Theorem A.1** (Theorem 4.9 from [Freedman, 2017]). *If $P$ is ergodic, with stationary $\pi$, then there exist constants $0 < \alpha < 1$ and $C > 0$ such that:*

$$\max_{v \in V} ||P^t(v, \cdot) - \pi||_{TV} \leq C \cdot \alpha^t \tag{6}$$

*More specifically, let $r$ be the smallest integer for which $P^r_{uv} > 0$ for all $u, v \in V$ and $\theta = 1 - \min_{u,v \in V} \frac{P^r_{uv}}{\pi(v)}$. Then, the constants $\alpha, C$ above are defined as:*

$$\alpha := \theta^{1/r} \quad and \quad C = \frac{1}{\theta}$$

*We call $\alpha$ and $C$ the **convergence parameters** of the Markov Chain.*

The following theorem is a Chernoff-Hoeffding bound for Discrete-Time Markov Chains:

**Theorem A.2** (Theorem 3.1. from [Chung et al., 2012]). *Let $M$ be an ergodic Markov Chain with state space $[n]$ and stationary distribution $\pi$. Let*

$$t_{\mathrm{mix}} = t_{\mathrm{mix}}(\varepsilon) = \min\{t : \max_{x \in \Delta_n} ||xM^t - \pi||_{TV} \leq \varepsilon\}$$

*be its $\varepsilon$-mixing time for $\varepsilon \leq 1/8$. Let $(V_1, ..., V_t)$ denote a $t$-step random walk on $M$ starting from an initial distribution $\phi$ on $[n]$, i.e. $V_1 \to \phi$. For every $i \in [t]$, let $f_i : [n] \to [0, 1]$ be a weight function at step $i$ such that $\mathbb{E}_{v \leftarrow \pi}[f_i(v)] = \mu$ for all $i$. Define the total weight of the walk by:*

$$X := \sum_{i=1}^{t} f_i(V_i)$$

*There exists some constant $c$ such that:*

$$\Pr[X \geq (1 + \delta)\mu t] \leq \begin{cases} c||\phi||_\pi \cdot \exp(-\delta^2 \mu t/(72T)), & for \ 0 \leq \delta \leq 1 \\ c||\phi||_\pi \cdot \exp(-\delta \mu t/(72T)), & for \ \delta > 1 \end{cases} \tag{7}$$

$$\Pr[X \leq (1 - \delta)\mu t] \leq c||\phi||_\pi \exp(-\delta^2 \mu t/(72T)), \ for \ 0 \leq \delta \leq 1 \tag{8}$$

*where $|| \cdot ||_\pi$ is the $\pi$-norm induced by the inner product under the $\pi$-kernel:*

$$\langle u, v \rangle_\pi = \sum_{i \in [n]} \frac{u_i v_i}{\pi(i)}$$

# B    Proof of Lemma 3.1

**Lemma B.1.** *Let $(X_t)_{t \geq 0}$ and $(Y_t)_{t \geq 0}$ be two random walks starting at states $X_0 = u$ and $Y_0 = v$. Let $T = \min\{t \geq 0 : X_t = Y_t\}$ be the meeting time of the two random walks. Then,*

$$\Pr[T > t] \leq O\left(\frac{1}{\sqrt{\pi_G(u)\pi_G(v)}} \exp\left(-\frac{||\pi_G||_2^2 t}{72 t_{\mathrm{mix}}}\right)\right)$$

*Proof.* Note that $Z_t = (X_t, Y_t) \in V \times V$ can be modeled as a random walk on the Kronecker product $G \times G$, where $\pi_{G \times G}(x, y) = \pi(x)\pi(v)$ and $t_{\mathrm{mix}}(G \times G) = t_{\mathrm{mix}}(G)$. In order to apply a

Markov Chain Chernoff bound (Theorem A.2), we define the weight function $f(x, y) = 1$ if $x = y$ and $f(x, y) = 0$, otherwise. Let $F = \sum_{\tau=0}^{t} f(X_i, Y_i)$ and observe that $t < T$ if and only if $F = 0$. Since $\pi_{G \times G}(x, y) = \pi_G(x)\pi_G(y)$ we also have $\mu = \|\pi_G\|_2^2$. If $\delta = 1$, then Theorem A.2 gives:

$$\Pr[T > t] = \Pr[F = 0] = \Pr[F \geq (1 - \delta)\mu t] \leq O\left(\frac{1}{\sqrt{\pi_G(u)\pi_G(v)}} \exp\left(-\frac{\|\pi_G\|_2^2 t}{72 t_{\text{mix}}}\right)\right).$$

$\square$

## C    Proof of Theorem 4.1

In this Section we prove Theorem 4.1.

**Theorem C.1.** *Algorithm 3 returns an estimate $\widehat{h}_G(s, t)$ such that with probability at least $9/10$ it holds that:*

$$\left|\widehat{h}_G(s, t) - H_G(s, t)\right| \leq \varepsilon$$

*Its runtime is bounded by*

$$O\left(\frac{\ell^4}{\varepsilon^2 \cdot \pi_G(v)^2} \log \ell\right), \quad \text{where} \quad \ell = \frac{\log \frac{n}{\varepsilon - \varepsilon\lambda}}{\log \frac{1}{\lambda}}$$

*for the spectral gap $\lambda := \max\{|\lambda_2|, |\lambda_n|\}$ of $G$.*

*Proof.* It is easy to see that the algorithm runs in $\widetilde{O}(r\ell^2)$, which gives us the claimed runtime. We therefore just need to argue about the algorithm's correctness. Analogously to the effective resistance approach taken in [Peng et al., 2021], we want to find a cut-off point for the series in Equation 4. That is, we want to find as small an $\ell(\varepsilon)$ as possible such that:

$$\left|H_G(u, v) - \sum_{i=0}^{\ell-1}\left(\mathbf{1} - \frac{1}{\pi(v)}\mathbf{1}_v\right)^\top W^i \chi_{uv}\right| \leq \frac{\varepsilon}{2}$$

We first spectrally decompose the $i$-th power of $W$:

$$W^i = \sum_{j=1}^{n} \lambda_j^i w_j w_j^T$$

as before. Then, we have that:

$$\left|\sum_{i=\ell}^{\infty}\left(\mathbf{1} - \frac{1}{\pi(v)}\mathbf{1}_v\right)^\top W^i \chi_{uv}\right|$$

$$= \left|\sum_{i=\ell}^{\infty}\left(\mathbf{1} - \frac{1}{\pi(v)}\mathbf{1}_v\right)^\top \left(\sum_{j=1}^{n} \lambda_j^i w_j w_j^T\right)\chi_{uv}\right|$$

$$= \left|\sum_{i=\ell}^{\infty}\sum_{j=2}^{n}\left(\mathbf{1} - \frac{1}{\pi(v)}\mathbf{1}_v\right)^\top \left(\lambda_j^i w_j w_j^T\right)\chi_{uv}\right| \qquad \text{(as } w_1^T \chi_{st} = 0)$$

$$\leq \left|\sum_{i=\ell}^{\infty}\lambda^i \sum_{j=2}^{n}\left(\mathbf{1} - \frac{1}{\pi(v)}\mathbf{1}_v\right)^\top \left(w_j w_j^T\right)\chi_{uv}\right| \qquad \text{(Definition of } \lambda)$$

We give a straightforward bound for the main term. If $j \in \{2, ..., n\}$:

$$\left(\mathbf{1} - \frac{1}{\pi(v)}\mathbf{1}_v\right)^\top \left(w_j w_j^T\right)\chi_{uv} \leq ||w_j w_j^T \chi_{uv}||_1 \leq \sqrt{n}$$

since we can choose $w_j$ so that $||w_j||_2 = 1$. Thus:

$$\left| H_G(s,t) - \sum_{i=0}^{\ell-1} \left( \mathbf{1} - \frac{1}{\pi(t)} \mathbf{1}_t \right)^\top W^i \chi_{st} \right| \le \left| \sum_{i=\ell}^{\infty} \lambda^i n\sqrt{n} \right| \le \frac{\lambda^\ell}{1-\lambda} \cdot n\sqrt{n}$$

since $\lambda < 1$. Now, we aim to set $\lambda$ such that:

$$\frac{\lambda^\ell}{1-\lambda} \cdot n\sqrt{n} \le \frac{\varepsilon}{2}$$

It suffices, thus, to set our threshold to:

$$\ell = \frac{\log \left( \frac{4n\sqrt{n}}{\varepsilon - \varepsilon\lambda} \right)}{\log(1/\lambda)} = O\left( \frac{\log \frac{n}{\varepsilon - \varepsilon\lambda}}{\log(1/\lambda)} \right)$$

Therefore we only need to ensure that:

$$\left| \widehat{h}_G(u,v) - \sum_{i=0}^{\ell-1} \left( \mathbf{1} - \frac{1}{\pi(v)} \mathbf{1}_v \right)^\top W^i \chi_{uv} \right| \le \frac{\varepsilon}{2} \tag{9}$$

and then by triangle inequality we would be done. Our algorithm approximates, for each $i \in \{0,1,...,\ell-1\}$ the following term:

$$\left( \mathbf{1} - \frac{1}{\pi(v)} \mathbf{1}_v \right)^\top W^i \chi_{uv} = \cancel{\mathbf{1}^\top W^i \mathbf{1}_u} - \frac{1}{\pi(v)} \mathbf{1}_v^\top W^i \mathbf{1}_u - \cancel{\mathbf{1}^\top W^i \mathbf{1}_v} + \frac{1}{\pi(v)} \mathbf{1}_v^\top W^i \mathbf{1}_v$$

$$= \frac{1}{\pi(v)} \left( \mathbf{1}_v^\top W^i \mathbf{1}_v - \mathbf{1}_v^\top W^i \mathbf{1}_u \right)$$

where the cancellations in the first equality follow from the fact that $\mathbf{1}^\top W^i = \mathbf{1}^\top$. Now we have that:

$$\mathbf{1}_v^\top W^i \mathbf{1}_v = \Pr[v \to^i v] \quad \text{and} \quad \mathbf{1}_v^\top W^i \mathbf{1}_u = \Pr[v \to^i u]$$

where $\Pr[s \to^i t]$ is the probability a random walk starting from $s$ reaches $t$ in $i$ steps. We estimate these probabilities by performing $r$ independent random walks starting at $v$ and letting $T_u$ and $T_v$ be the number of those walks ending up in $u$ and $v$ respectively. Then, we estimate:

$$\widehat{p_{i,u}} = \frac{2m}{\deg(v)} \cdot \frac{T_u}{r} \quad \text{and} \quad \widehat{p_{i,v}} = \frac{2m}{\deg(v)} \cdot \frac{T_v}{r}$$

We know that as a sum of indicators we have:

$$\mathbb{E}[T_s] = r \cdot \Pr[t \to^i s]$$

and so the Hoeffding bound gives:

$$\Pr\left[ \left| \widehat{p_{i,s}} - \frac{1}{\pi(t)} \Pr[t \to^i s] \right| \ge \frac{\varepsilon}{4\ell} \right] = \Pr\left[ |T_s - \mathbb{E}[T_s]| \ge r \cdot \frac{\varepsilon \deg(t)}{8\ell m} \right]$$

$$\le 2\exp\left( \frac{-2\varepsilon^2 \pi(v)^2}{64\ell^2} \cdot r \right)$$

$$\le \frac{1}{20\ell}$$

when we set:

$$r \ge \frac{32\ell^2}{\pi(v)^2 \varepsilon^2} \log(40\ell)$$

Through a simple use of the union bound over $2\ell$ sums, we conclude that Equation 9 holds with probability at least $9/10$, which concludes our theorem. $\qquad\square$

# D  A Walk Sampling Algorithm

In this section we analyze a simple algorithm for estimating the hitting time $H_G(s,t)$: sampling a collection of random walks from $s$ of various lengths and counting how many of them hit $t$. The analysis uses a Chernoff bound for Markov Chains [Chung et al., 2012].

**Theorem D.1.** *Let $M$ be an ergodic Markov chain with stationary distribution $\pi$. Let $s$ and $t$ be two different states. There exists an algorithm calculating the hitting time $H_G(s,t)$ to within an absolute error of $\varepsilon$ with probability at least $\frac{1}{n}$ that uses*

$$\widetilde{O}\left(\frac{T^2}{\pi(t)^2\varepsilon^2}\right)$$

*random walk samples of length at most $\widetilde{O}(T/\pi(t))$, where $T$ is the $\varepsilon$-mixing time of $M$.*

*Proof.* Recall the definition of the hitting time:

$$H_{st} = \sum_{i=0}^{\infty} i \cdot \Pr\left[T_1(t) = i \mid X_0 = s)\right],$$

We wish to identify a threshold $\ell$ such that:

$$\sum_{i=l}^{\infty} i \cdot \Pr\left[T_1(t) = i \mid X_0 = s)\right] < \varepsilon$$

In the context of Theorem A.2, let $f(u) = 1$ if $u = t$ and $f(u) = 0$ otherwise. Then, $\mu = \mathbb{E}_{X\sim\pi}[f(X)] = \Pr_{X\sim\pi}[X = t] = \pi(t)$. Suppose $F^{(i)} = \sum_{j\leq i} f(X_j)$. Our distribution $\phi$ has weight 0 for all points other than $s$, so $||\phi||_\pi = \sqrt{\langle\phi,\phi\rangle_\pi} = \sqrt{\frac{1}{\pi(s)}}$ and thus we have:

$$\Pr[F^{(i)} \leq (1-\delta)\mu i] \leq \frac{c}{\sqrt{\pi(s)}}\exp\left(-\delta^2\frac{\mu i}{72T}\right) \iff$$

$$\Pr[F^{(i)} \leq 0] \leq \frac{c}{\sqrt{\pi(s)}}\exp\left(-\frac{\pi(t)i}{72T}\right)$$

where we chose $\delta = 1$. For all $i$, we have $\Pr[F^{(i)} = 0] \geq \Pr[T_1(t) = i + 1 \mid X_0 = s]$ and thus

$$\sum_{i=\ell-1}^{\infty}(i+1)\cdot\Pr[T_1(t) = (i+1) \mid X_0 = s] \leq \sum_{i=\ell-1}^{\infty}(i+1)\cdot\Pr[F^{(i)} = 0]$$

$$\leq \frac{c}{\sqrt{\pi(s)}}\sum_{i=\ell-1}^{\infty}(i+1)\cdot\exp\left(-\frac{\pi(t)i}{72T}\right)$$

$$= \frac{c}{\sqrt{\pi(s)}}\sum_{i=\ell-1}^{\infty}(i+1)\cdot\alpha^i$$

$$= \frac{c}{\sqrt{\pi(s)}}\left(\frac{\alpha^{\ell-1}}{(1-\alpha)^2} + \frac{(\ell-1)\alpha^{\ell-1}}{1-\alpha}\right)$$

$$\leq \frac{2c\ell}{\sqrt{\pi(s)}}\cdot\frac{\alpha^{\ell-1}}{1-\alpha}$$

for $\alpha = \exp(-\pi(t)/72T) \ll 1$, where for the derivation we used differentiation on the geometric series. We seek some $\ell$ that makes the tail have weight at most $\frac{\varepsilon}{r\ell}$, for reasons we will discuss shortly, where $r = \ell^2/\varepsilon^2$:

$$\frac{2c\ell}{\sqrt{\pi(s)}}\cdot\frac{\alpha^{\ell-1}}{1-\alpha} \leq \frac{\varepsilon}{r\ell} \iff \ell^4\alpha^{\ell-1} \leq \frac{\varepsilon^3(1-\alpha)\sqrt{\pi(s)}}{2c}$$

$$(\ell-1)\ln(\alpha) + 4\ln(\ell) \leq \ln\frac{\varepsilon^3(1-\alpha)\sqrt{\pi(s)}}{2c}$$

$$\ell \geq 1 + \frac{1}{\ln \alpha} \left( \ln \frac{\varepsilon^3 (1-\alpha) \sqrt{\pi(s)}}{2c} - 4 \ln \ell \right)$$

Since we must have that $\ell \geq 1$, it suffices to set:

$$\ell \geq \max \left\{ 1, 1 + \frac{\ln \frac{\varepsilon^3 (1-\alpha) \sqrt{\pi(s)}}{2c}}{\ln \alpha} \right\} = \Omega \left( \frac{T}{\pi(t)} \ln \frac{\sqrt{\pi(s)}}{\varepsilon^3} \right)$$

This means that it suffices to take random walks of length approximately equal to the $\varepsilon$-mixing time. Next, we need to figure out how many of those random walks we need.

Assume now we perform $r$ independent random walks from $s$ to $t$ and record the number of steps as the random variables $X_1, \ldots, X_r$. Let us use $E$ to denote the event that $X_i \leq \ell$ for all $i$. By another application of Theorem A.2 and our choice of $\ell$,

$$\Pr[X_i > \ell] = \Pr[F^{(\ell)} \leq 0] \leq \frac{c}{\sqrt{\pi(s)}} \exp \left( -\frac{\pi(t)\ell}{72T} \right) \leq \frac{\varepsilon}{r\ell}$$

and thus $\Pr[\overline{E}] \leq r \cdot \frac{\varepsilon}{r\ell} = \frac{\varepsilon}{\ell}$ by a union bound. By the law of total expectation,

$$\left| \mathbb{E} \left[ \frac{1}{r} \sum_{i=1}^{r} X_i \right] - \mathbb{E} \left[ \frac{1}{r} \sum_{i=1}^{r} X_i \middle| E \right] \right| = |\mathbb{E}[X_1] - \mathbb{E}[X_1 \mid E]|$$

$$= \Pr[\overline{E}] \cdot \left| \mathbb{E}[X_1 \mid E] - \mathbb{E}[X_1 \middle| \overline{E}] \right|$$

$$\leq \ell \Pr[\overline{E}] + \Pr[\overline{E}] \mathbb{E}[X_1 \mid \overline{E}]$$

$$= \ell \Pr[\overline{E}] + \sum_{i=\ell+1}^{\infty} i \cdot \Pr[X_1 = i \cap \overline{E}]$$

$$= \ell \Pr[\overline{E}] + \underbrace{\sum_{i=\ell+1}^{\infty} i \cdot \Pr[X_1 = i]}_{\leq \varepsilon} \quad (*)$$

$$\leq 2\varepsilon$$

On the other hand, by a Hoeffding bound taken after conditioning on $E$, we have that:

$$\Pr \left[ \left| \frac{1}{r} \sum_{i=1}^{r} X_i - \mathbb{E} \left[ \frac{1}{r} \sum_{i=1}^{r} X_i \middle| E \right] \right| \geq \varepsilon \middle| E \right] \leq 2 \exp \left( -\frac{2r\varepsilon^2}{\ell^2} \right)$$

If we choose

$$r = O \left( \frac{\ell^2 \lg n}{\varepsilon^2} \right) = \tilde{O} \left( \frac{T^2}{\pi(t)^2 \varepsilon^2} \right)$$

then, in combination with (*) we get that with probability at least $1 - \frac{1}{n}$:

$$\left| \frac{1}{r} \sum_{i=1}^{r} X_i - \mathbb{E} \left[ \frac{1}{r} \sum_{i=1}^{r} X_i \right] \right| \leq 3\varepsilon$$

and that concludes our proof. $\qquad\square$

## E  Lower Bound to the Number of Sampled Walks

A natural question is whether the upper bound established by our analysis of the walk sampling algorithm is tight. In this section we answer this question in the affirmative, showing that the number of random walk samples required to estimate the hitting time is $\Omega(\frac{t_{\mathrm{mix}}^2}{\pi(t)^2 \varepsilon^2})$ for a certain class of graphs. This result can be interpreted in the sense that there is no "free lunch" in estimating the hitting time of a Markov Chain, as there are always pathological graphs that require many samples.

Our lower bound applies to a class of "barbell" graphs with one star and one clique connected by a path. The main idea is that we can model the hitting time from the clique to the start by the product $\Theta(n^2) \cdot \mathrm{Geom}(1/n)$. Arguing about the anti-concentration of geometric-like distributions we arrive at the desired result.

### E.1 Preliminaries

Our analysis starts with the following standard anti-concentration bounds on the Binomial Distribution:

**Lemma E.1** (Anti-concentration of upper tail, [Mousavi, 2010]). *Let $X_1, ..., X_m$ be iid Bernoulli random variables with parameter $p$. If $p \leq 1/4$, then for any $t \geq 0$ we have:*

$$\Pr\left[\sum_{i=1}^{m} X_i \geq mp + t\right] \geq \frac{1}{4} \exp\left(-\frac{2t^2}{mp}\right) \tag{10}$$

We use this result to give a bound on the lower tail of the geometric distribution:

**Lemma E.2.** *Let $Y_1, Y_2, \ldots, Y_k \sim \mathrm{Geom}(1/\mu)$ and $Y = \sum_{i=1}^{k} Y_i$. Let $\overline{Y} = \frac{1}{k}Y$. Then if $\varepsilon$ is a positive constant with $\varepsilon \leq \mu/2$, we have:*

$$\Pr\left[\overline{Y} \leq \mu - \varepsilon\right] \geq \frac{1}{4} \exp\left(-\frac{4k\varepsilon^2}{\mu^2}\right)$$

*Proof.* Let us denote with $\mathrm{Bin}(r)$ the sum of $r$ Bernoulli random variables with probability $1/\mu$. The event of having $Y \leq k(\mu - \varepsilon)$ corresponds to performing at most $k(\mu - \varepsilon)$ such Bernoulli trials before observing $k$ successes. Alternatively, the number of successes in the first $k(\mu - \varepsilon)$ Bernoulli trials is at least $k$. So:

$$\Pr\left[\overline{Y} \leq \mu - \varepsilon\right] = \Pr\left[\mathrm{Bin}(k(\mu - \varepsilon)) \geq k\right]$$

Let $\nu_- = \mathbb{E}[\mathrm{Bin}(k(\mu - \varepsilon))] = k - k\frac{\varepsilon}{\mu}$. Since $\mu \geq 2\varepsilon$, we have that $\mu^2 - \varepsilon\mu \geq \frac{\mu^2}{2}$, and so by our upper tail anti-concentration bound in Lemma E.1 we have:

$$\begin{aligned}
\Pr\left[\mathrm{Bin}(k(\mu - \varepsilon)) \geq k\right] = \Pr&\left[\mathrm{Bin}(k(\mu - \varepsilon)) \geq \nu_- + k\frac{\varepsilon}{\mu}\right] \\
&\geq \frac{1}{4} \exp\left(-\frac{2k^2\varepsilon^2}{\mu^2} \cdot \frac{1}{k - k\varepsilon/\mu}\right) \\
&= \frac{1}{4} \exp\left(-\frac{2k\varepsilon^2}{\mu^2 - \varepsilon\mu}\right) \\
&\geq \frac{1}{4} \exp\left(-\frac{4k\varepsilon^2}{\mu^2}\right)
\end{aligned}$$

Therefore, we get that:

$$\Pr\left[|\overline{Y} - \mu| \geq \varepsilon\right] \geq \frac{1}{4} \exp\left(-\frac{4k\varepsilon^2}{\mu^2}\right)$$

as desired. $\qquad\square$

We will also use the following "relaxation lemma" to decouple a random process into two easier-to-analyze processes and lower bound each anti-concentration separately.

**Lemma E.3** (Anti-Concentration Relaxation). *Let $\mathcal{T}$ be a distribution over $\mathbb{N}_{>0}$ with mean $\mu_{\mathcal{T}} \geq 1$ and $\mathcal{A}$ be a right skewed distribution over $\mathbb{R}_{\geq 0}$ with mean $\mu_{\mathcal{A}} > 0$. Further, let $T \sim \mathcal{T}$ and $A_1, A_2, \cdots \sim \mathcal{A}$ be independent samples. Define the random variables $X = \sum_{t=1}^{T} A_t$ and $Y = \mu_{\mathcal{A}} T$ with means $\mu = \mathbb{E}[X] = \mathbb{E}[Y] = \mu_{\mathcal{T}}\mu_{\mathcal{A}}$. Then we have:*

$$\Pr\left[X \leq \mu - \varepsilon\right] \geq \frac{1}{2} \Pr\left[Y \leq \mu - \varepsilon\right]$$

*Proof.* Let $X := \sum_{t=1}^{T} A_t$, where $T \sim \mathcal{T}$ and $A_t \sim \mathcal{A}$ are all mutually independent random variables. We have the following sequence of bounds:

$$\Pr\left[X \leq \mu - \varepsilon\right] \geq \Pr\left[T \leq \mu_{\mathcal{T}} - \frac{\varepsilon}{\mu_{\mathcal{A}}} \wedge X \leq \mu - \varepsilon\right]$$

$$= \Pr\left[T \le \mu_{\mathcal{T}} - \frac{\varepsilon}{\mu_{\mathcal{A}}} \wedge \sum_{t=1}^{T} A_t \le \mu - \varepsilon\right]$$

$$\ge \Pr\left[T \le \mu_{\mathcal{T}} - \frac{\varepsilon}{\mu_{\mathcal{A}}} \wedge \sum_{t=1}^{\mu_{\mathcal{T}} - \frac{\varepsilon}{\mu_{\mathcal{A}}}} A_t \le \mu - \varepsilon\right] \qquad \text{(because } T \le \mu_{\mathcal{T}}\text{)}$$

$$= \Pr\left[T \le \mu_{\mathcal{T}} - \frac{\varepsilon}{\mu_{\mathcal{A}}}\right] \cdot \Pr\left[\sum_{t=1}^{\mu_{\mathcal{T}} - \frac{\varepsilon}{\mu_{\mathcal{A}}}} A_t \le \mu - \varepsilon\right] \qquad \text{(independence)}$$

Finally, because $\mathcal{A}$ is right-skewed, we can write:

$$\Pr\left[X \le \mu - \varepsilon\right] \ge \Pr[Y \le \mu - \varepsilon] \cdot \Pr\left[\sum_{t=1}^{\mu_{\mathcal{T}} - \frac{\varepsilon}{\mu_{\mathcal{A}}}} A_t \le \mu_{\mathcal{A}}\left(\mu_{\mathcal{T}} - \frac{\varepsilon}{\mu_{\mathcal{A}}}\right)\right]$$

$$= \Pr[Y \le \mu - \varepsilon] \cdot \Pr\left[\sum_{t=1}^{\mu_{\mathcal{T}} - \frac{\varepsilon}{\mu_{\mathcal{A}}}} A_t \le \mathbb{E}\left[\sum_{t=1}^{\mu_{\mathcal{T}} - \frac{\varepsilon}{\mu_{\mathcal{A}}}} A_t\right]\right]$$

$$\ge \frac{1}{2}\Pr[Y \le \mu - \varepsilon]$$

as desired. $\qquad\square$

As a corollary, consider applying the relaxation Lemma E.3 to the setting of an average of $r$ random variables $X_i$. We can change the definition of $\mathcal{T}$ to relax the anti-concentration in that setting as well:

**Corollary E.1.** *Let $\mathcal{T}$ be a distribution over $\mathbb{N}_{>0}$ with mean $\mu_{\mathcal{T}} \ge 1$ and $\mathcal{A}$ be a real, right-skewed distribution with mean $\mu_{\mathcal{A}} > 0$. Further, let $T \sim \mathcal{T}$ and $A_1, A_2, \cdots \sim \mathcal{A}$ be independent samples. Define the random variables $X = \sum_{t=1}^{T} A_t$ and $Y = \mu_{\mathcal{A}}T$ with means $\mu = \mathbb{E}[X] = \mathbb{E}[Y] = \mu_{\mathcal{T}}\mu_{\mathcal{A}}$. Then, suppose we have $r$ random variables $X_1, ..., X_r$, all independently and identically distributed. We have:*

$$\Pr\left[\frac{1}{r}\sum_{i=1}^{r} X_i \le \mu - \varepsilon\right] \ge \frac{1}{2}\Pr\left[\frac{1}{r}\sum_{i=1}^{r} Y_i \le \mu - \varepsilon\right]$$

*where $Y_1, ..., Y_r$ are independently defined as $Y_i \sim \mu_{\mathcal{A}} \cdot \mathcal{T}$.*

*Proof.* Consider a random variable $T' = \sum_{i=1}^{r} T_i$ where $T_i \sim \mathcal{T}$. Suppose $T' \sim \mathcal{T}'$. Then:

$$\frac{1}{r}\sum_{i=1}^{r} X_i \equiv \sum_{i=1}^{T'} A_i'$$

where $A_i' \sim \frac{1}{r}\mathcal{A}$ is right skewed as scaling by a positive value does not affect skewness. Since $\mu_{\mathcal{T}'} = r\mu_{\mathcal{T}}$ and $\mu_{\mathcal{A}'} = \frac{1}{r}\mu_{\mathcal{A}}$, we can apply Lemma E.3 to get:

$$\Pr\left[\frac{1}{r}\sum_{i=1}^{r} X_i \le \mu - \varepsilon\right] \ge \frac{1}{2}\Pr\left[\frac{1}{r}\sum_{i=1}^{r} Y_i \le \mu - \varepsilon\right]$$

$\qquad\square$

### E.2 The Lower Bound Proof

We now proceed to proving our lower bound.

**Theorem E.1.** *Let $G$ be a graph and $u, v$ be two different vertices of it. Suppose an algorithm is able to estimate $H_G(u, v)$ within additive error $\varepsilon$ with constant probability of success by taking the average over $r$ random walk samples of unbounded length. Then we must have that:*

$$r = \Omega\left(\frac{t_{\mathrm{mix}}^2}{\pi(v)^2\varepsilon^2}\right)$$

*in the worst case, where $t_{\mathrm{mix}}$ is the mixing time of the graph.*

*Proof.* Let $G$ be a "barbell"-like graph, as in Figure 3, consisting of a cliques $Y = \{y_1, ..., y_n\}$ and a star $S = \{v_1, ..., v_{n^2}, x\}$ connected via a path $P := (x = u_n \leftrightarrow u_{n-1} \leftrightarrow \cdots \leftrightarrow u_1 \leftrightarrow y_1)$

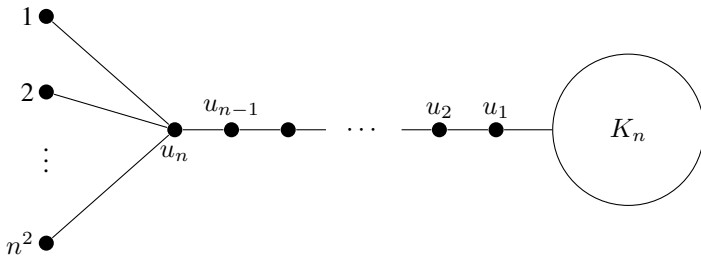

Figure 3: A 'barbell'-like graph: a star, a path and a clique

Suppose we want to estimate $H_G(u_1, u_n)$. Our estimation procedure samples $r$ random walks from $u_1$ and averages the time they take to reach $u_n$. Let $t_i$ be the time it takes for the $i$-th random walk to reach $u_n$. Then our estimator:

$$\text{est} := \frac{1}{r} \sum_{k=1}^{r} t_k$$

is an unbiased estimator of the hitting time.

We claim that $r = \Omega(n^6/\varepsilon^2)$ samples are required to guarantee that:

$$|\text{est} - H_G(u_1, u_n)| \leq \varepsilon$$

By a Theorem of Feige [Feige, 1995] for the hitting time in lollipop graphs, we know that the maximum hitting time from any vertex in $P \cup Y$ to any other vertex is $\Theta(n^3)$. If the random walk starts from some vertex in $S$, then it takes it an expected $\Theta(n^2)$ steps until it reaches $u_{n-1}$. By a standard "drunkard's walk" argument we conclude that the expected hitting time to any vertex in $Y$ is $\Theta(n^3)$. Therefore, by a Theorem of Aldous [Aldous and Fill, 2002, Aldous, 1982] stating that the mixing time is characterized by the maximum hitting time, we get that $t_{\text{mix}} = \Theta(n^3)$. We also know that $\pi(u_n) = \Theta(1)$, and therefore the $\Omega(n^6/\varepsilon^2)$ lower bound gives us the desired $\Omega(t_{\text{mix}}^2/(\pi(v)^2\varepsilon^2))$ lower bound.

For the remainder of the proof we focus on proving that $r \geq \Omega(n^6/\varepsilon^2)$ is required. Our proof relies on modeling the random walk from $u_1$ to $u_n$ through a chain of independent samples from appropriate distributions. Consider a random walk starting at $u_1$ and let $p$ be the probability that it reaches $u_n$ *before entering the clique* $Y$. It is a standard calculation that $p = 1/n$: Let $p_i$ be the probability a random walk reaches $u_1$ before entering $Y$ if it starts from $u_i$. By conditioning on the first step of the random walk, we get the recurrence:

$$p_i = \frac{1}{2}(p_{i-1} + p_{i+1})$$

with boundary conditions $p_1 = 1$ and $p_{n+1} = 0$. We solve this by $p_i = \frac{n-i+1}{n}$, which gives the expression $p = p_n = \frac{1}{n}$. Starting from $u_1$ and conditioned on the event that it does not reach $u_n$, the random walk can either follow the path, or become trapped in the clique. As a result, we can model the time $t$ it takes for a random walk starting at $u_1$ to reach $u_n$ as:

$$t \sim \sum_{i=1}^{t_Y} A_i$$

where $t_Y \sim \text{Geom}(1/n)$ and:

$$A_i \sim \begin{cases} \text{return time to } u_1 \text{ in path } P, & \text{with probability } \frac{1}{2} \\ \text{return time to } u_1 \text{ from clique } Y, & \text{with probability } \frac{1}{2} \end{cases}$$

where the $A_i$ random variables are all mutually independent. Both return times in the distribution of $A_i$ can be modeled as geometric random variables. Thus, their mixture is right skewed, so we can invoke Corollary E.1 to get that:

$$\Pr\left[\frac{1}{r}\sum_{i=1}^{r} t_i \leq \mu - \varepsilon\right] \geq \frac{1}{2}\Pr\left[\frac{1}{r}\sum_{i=1}^{r} G_i \leq n - \frac{\varepsilon}{\Theta(n^2)}\right]$$

with $G_i \sim \mathrm{Geom}(1/n)$. By Lemma E.2 we have that

$$\Pr\left[\frac{1}{r}\sum_{i=1}^{r}G_i \leq n - \frac{\varepsilon}{\Theta(n^2)}\right] \geq \frac{1}{4}\exp\left(-\frac{4r\varepsilon^2}{n^6}\right)$$

Since we seek an algorithm that makes the error at most $\varepsilon$ with at least a constant probability, we must have that $r = \Omega(n^6/\varepsilon^2)$, as initially claimed. $\qquad\square$

# F   Local Algorithms via Mixing Time Testing

In this section we expand upon our previous discussion on using property testing for hitting time estimation.

## F.1   Choosing a cutoff using local mixing times

For simplicity, we focus on the calculation of effective resistances. Recall the following definition:

**Definition F.1** (Effective Resistance). *The effective resistance between vertices $s, t \in V$ is:*

$$R_G(s,t) = \chi_{s,t}^T L^+ \chi_{s,t} \tag{11}$$

*where the Laplacian is defined as $L := D - A$, with $D$ being the degree diagonal matrix and $A$ being the adjacency matrix.*

Yoshida and Peng [Peng et al., 2021] show that the effective resistance can be written as a series with a light tail.

**Lemma F.1** ([Peng et al., 2021]). *It holds that:*

$$R_G(s,t) = \chi_{st}^{\top}\sum_{i=0}^{\infty}P^i D^{-1}\chi_{st} \tag{12}$$

At this point, we depart from the exposition of [Peng et al., 2021] and bound the tail of this series by using a different quantity: the $\varepsilon$-$(s,t)$-mixing time, which we define as follows:

**Definition F.2.** *Let $s, t \in V$ and $\varepsilon > 0$. Then the $\varepsilon$-$(s,t)$-mixing time is defined as the minimum number of steps required so that random walks starting from $s$ and $t$ have roughly the same distribution over the states:*

$$t_{\min}^{(\varepsilon)}(s,t) := \min\{i \in \mathbb{N} \mid ||\mathbf{1}_s P^i - \mathbf{1}_t P^i||_1 \leq \varepsilon\} \tag{13}$$

We prove that the $\varepsilon$-$(s,t)$-mixing time can be used as the cutoff point instead of $\lambda$ for irreducible and aperiodic Markov Chains.

**Lemma F.2.** *Assume that our Markov chain is ergodic with convergence parameters $(\alpha, C)$. Let*

$$\ell := t_{\min}^{(\varepsilon/4)} + \frac{\log\left(\frac{C}{\varepsilon'(1-\alpha)}\right)}{-\log\alpha}$$

*Then it holds that:*

$$\left|R_G(s,t) - \sum_{i=0}^{\ell}\chi_{st}^{\top}P^i D^{-1}\chi_{st}\right| \leq \frac{\varepsilon}{2} \tag{14}$$

*Proof.* If we truncate the series at Equation 12 after index $\ell$, the error we sustain is at most:

$$\left|\sum_{i=\ell}^{\infty}\chi_{st}^{\top}P^i D^{-1}\chi_{st}\right| \leq \sum_{i=\ell}^{\infty}|\chi_{st}^{\top}P^i D^{-1}\chi_{st}|$$

Suppose $\varepsilon' = \varepsilon/4$. Now, we have by Hölder's inequality that:

$$|\chi_{st}^{\top}P^i D^{-1}\chi_{st}| \leq ||\chi_{st}^{\top}P^i||_1 \cdot ||D^{-1}\chi_{st}||_{\infty}$$

$$\leq ||\mathbf{1}_s P^i - \mathbf{1}_t P^i||_1 \cdot \left( \frac{1}{\min_{v \in V} \deg(v)} - \frac{1}{\max_{v \in V} \deg(v)} \right)$$

$$\leq \varepsilon'$$

for $\ell \geq t_{\min}^{(\varepsilon')}(s,t)$. We further claim that this term decreases at an exponential rate for ergodic Markov Chains.

**Claim F.1.** *It holds that $f(i) := |\chi_{st}^\top P^i D^{-1} \chi_{st}| = O(\alpha^i)$ for some constant $0 < \alpha < 1$.*

*Proof.* We have that:
$$f(i) := |x_{st}^\top P^i D^{-1} x_{st}| \leq |P_{ss}^i - 2P_{st}^i + P_{tt}^i|$$

Let $\eta = O(\alpha^t)$. Since our Markov Chain is ergodic, Theorem A.1 gives that for any $w \in V$ we have $P_{uw}^i = (P^i 1_u)_w \leq \pi_w - \eta$ and $P_{uw}^i \geq \pi_w - \eta$, so:

$$f(i) \leq |\pi_u + \eta - \pi_u - \pi_v + 2\eta + \pi_v + \eta| = 4\eta = O(\alpha^t)$$

$\square$

With Claim F.1 in place, we can bound the error term as follows:

$$\left| \sum_{i=\ell}^\infty \chi_{st}^\top P^i D^{-1} \chi_{st} \right| < \varepsilon' + C \sum_{i=\ell}^\infty \alpha^i \leq \varepsilon' + \frac{C\alpha^\ell}{1-\alpha}$$

Setting $\ell > \frac{\log\left(\frac{C}{\varepsilon'(1-\alpha)}\right)}{-\log \alpha}$ bounds the error by $2\varepsilon' = \frac{\varepsilon}{2}$. $\square$

**Remark F.1.** *In general, this approach suffers from the same "locality" issue as the original algorithm of [Peng et al., 2021] which requires knowing $\lambda$. In fact, the rate $\alpha$ is closely connected to $\lambda$ and they are both connected to the overall mixing time of the Markov chain via Cheeger's inequality. Even with a good estimate of $t_{\min}^{\varepsilon/4}$ we are unable to bound the error in really degenerate cases where $\alpha$ is very close to 1, if we do not know $\alpha$ itself, which, like $\lambda$, is a global property. Therefore, for the remainder of this section we shall assume that $\ell = \Theta(t_{\min^{\varepsilon/4}})$, ignoring such degenerate cases.*

## F.2   Guessing $t_{\min}^{(\varepsilon)}$ by property testing

Following Lemma F.2, we are finally ready to design a new algorithm for estimating $R_G(s,t)$, one that does not require knowledge of $\lambda$. The idea is to provide an upper bound for $t_{\min}^{(\varepsilon)}(s,t)$ by using a property tester. The algorithm of [Batu et al., 2013] can be used to decide whether a Markov chain is close to mixing or not. The notion of "almost-mixing" used by that algorithm relates to our localized mixing:

**Definition F.3.** *A Markov Chain is $(\varepsilon, t)$-mixing if there exists some distribution $s$ such that for all $u \in V$ we have:*
$$||\mathbf{1}_u P^t - s||_1 \leq \varepsilon$$

This kind of mixing implies $\varepsilon$-$(s,t)$-mixing;

**Lemma F.3.** *If a Markov chain is $(\varepsilon, t)$-mixing, then for any states $u, v \in V$ we have $t_{\min}^{(2\varepsilon)}(s,t) \leq t$*

*Proof.* Suppose $q$ is the distribution that is referenced by the $(\varepsilon, t)$-mixing definition. By the triangle inequality:
$$||\mathbf{1}_s P^t - \mathbf{1}_v P^t||_1 \leq ||\mathbf{1}_s P^t - q||_1 + ||\mathbf{1}_t P^t - q||_1 \leq 2\varepsilon$$

$\square$

As a result, if we can find some $t$ such that our Markov Chain is $(\varepsilon/2, t)$-mixing, we would have an upper bound to $t_{\min}^{(\varepsilon)}$. We can do this via binary search, assuming that we possess an oracle for deciding approximate mixing. The problem of deciding whether a Markov Chain is $(\varepsilon, t)$-mixing can be reduced to the problem of testing whether two distributions are close to each other in $\ell_1$-distance. The algorithm provided by [Batu et al., 2013] provides the following guarantee:

**Theorem F.1** (Mixing Test [Batu et al., 2013]). *Let $M$ be a Markov chain. Suppose we are given a tester $\mathcal{T}$ for closeness of distributions in $\ell_1$ norm with time complexity $T(n, \varepsilon, \delta)$ and distance gap $f(\varepsilon)$ that takes as input sample oracles to distributions $p, q \in \Delta([n])$ and outputs, with probability at least $1 - \delta$:*

- *"accept" if $||p - q||_1 \leq f(\varepsilon)$*

- *"reject" if $||p - q||_1 > \varepsilon$*

*Then there exists a tester $\mathcal{T}_{mixing}$ with time complexity $O(nt \cdot T(n, \varepsilon, \delta/n))$ such that:*

- *If $M$ is $(f(\varepsilon)/2, t)$-mixing, then $\Pr[M \text{ is accepted}] > 1 - \delta$,*

- *If $M$ is not $(\varepsilon, t)$-mixing, then $\Pr[M \text{ is accepted}] < \delta$.*

We can use the $\ell_1$ closeness tester of [Chan et al., 2014] as the tester $\mathcal{T}$. As shown in [Chan et al., 2014], this is an optimal sample complexity for any tester.

**Theorem F.2** (Closeness test [Chan et al., 2014]). *There exists a tester that runs in $O(\max\{n^{2/3}\varepsilon^{-4/3}, n^{1/2}\varepsilon^{-2}\})$ time that, with probability at least $2/3$, distinguishes between $p = q$ and $||p - q||_1 \geq \varepsilon$.*

Since $f(\varepsilon) = 0$, combining Theorem F.1 and Theorem F.2, along with a repetitive boosting argument gives us an algorithm $\mathcal{T}_{mixing}$ that runs in $O(t \cdot n^{5/3}\varepsilon^{-2} \log(1/\delta))$ time and can, with probability at least $1 - \delta$ distinguish whether a markov chain is $(\varepsilon, t)$-mixing or not.

Putting everything together, we consider *binary searching* for $t$ such that $M$ is $(\varepsilon, t)$-mixing. We know that $1 \leq t \leq D$, where $D$ is the diameter of the graph. Using $\mathcal{T}_{mixing}$ with a boosted success probability of $1 - \frac{\delta}{\log(D)}$ we conclude the following theorem:

**Theorem F.3.** *There exists a (local) algorithm that determines the smallest $t$ such that a Markov Chain $M$ with $n$ states is $(\varepsilon, t)$-mixing. The algorithm runs in $O(D \log(D) n^{5/3} \varepsilon^{-2} \log(1/\delta))$ time, where $D$ is the diameter of the graph.*

As a consequence, this algorithm gives an upper bound to $t_{\min}^{(\varepsilon)}$ and can therefore be used in the context of Lemma F.2 to approximate the effective resistance without knowledge of spectral gap $\lambda$.

# G    Additional Experimental Results

In this section we deposit additional experimental results and information pertaining to our algorithms.

## G.1    Synthetic Experiments Setup Details

Table 2 contains information about the networks we used for our synthetic experiments in the main body of the paper.

Table 2: Synthetic Networks

| Name | $n$ | $m$ |
|---|---|---|
| Erdos-Renyi $p = 0.01$ | 1 000 | $5\,007.4 \pm 32.0$ |
| Barabasi-Albert $k = 10$ | 1 000 | 9 900 |
| Communities $p_{\text{inter}} = 0.01, p_{\text{intra}} = 0.05$ | 1 000 | $9\,021.4 \pm 51.8$ |
| Football | 115 | 613 |
| Facebook | 4 039 | 88 234 |

## G.2    Benefits from parallelism

Our algorithms are highly parallelizable because their local nature lends itself to multi-threaded computation. Through parallelization we obtain even better performance in computing the hitting time compared to the exact solver, as shown in Table 3. Here, we compute hitting times on the Facebook network using 10000 random walks.

Table 3: Speedup through parallelism. We show the running time in seconds.

| Algorithm / Number of cores | 1 | 5 | 10 | 20 |
|---|---|---|---|---|
| Meeting Time Algorithm | $0.554 \pm 0.071$ | $0.377 \pm 0.065$ | $0.335 \pm 0.026$ | $0.359 \pm 0.018$ |
| Exact Algorithm | $2.165 \pm 1.270$ | $3.556 \pm 2.047$ | $1.255 \pm 0.026$ | $1.238 \pm 0.024$ |

## G.3 Ablation studies

We present additional ablation studies evaluating runtime and relative estimation error as functions of the number of random walks, using the Football and Facebook networks. We also perform the same experiment for our synthetic datasets. In the Football network, our cutoff algorithm under-performs the others, likely due to an overestimated $\lambda$ parameter—highlighting its sensitivity to hyperparameter tuning. In contrast, Algorithm 1 appears more robust. We also visualize the correlation between hitting times and two pair sampling measures, illustrating how our method differs from simpler uniform sampling.

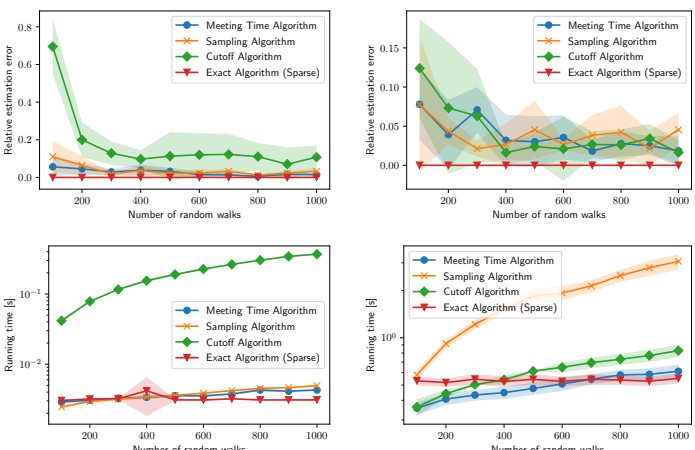

Figure 4: Ablation study for the number of random walks in the Football and Facebook networks.

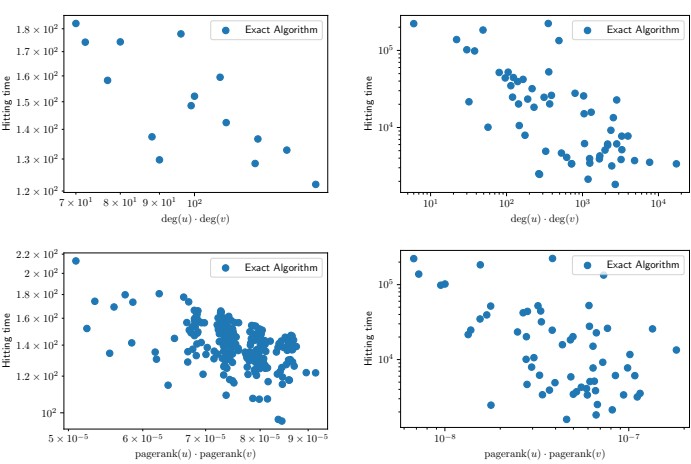

Figure 5: Correlation between the hitting time and the degree product $\deg(u) \cdot \deg(v)$ (top) and the product of pagerank centralities $\mathrm{pagerank}(u) \cdot \mathrm{pagerank}(v)$ (bottom) on the Football (left) and Facebook (right) networks.

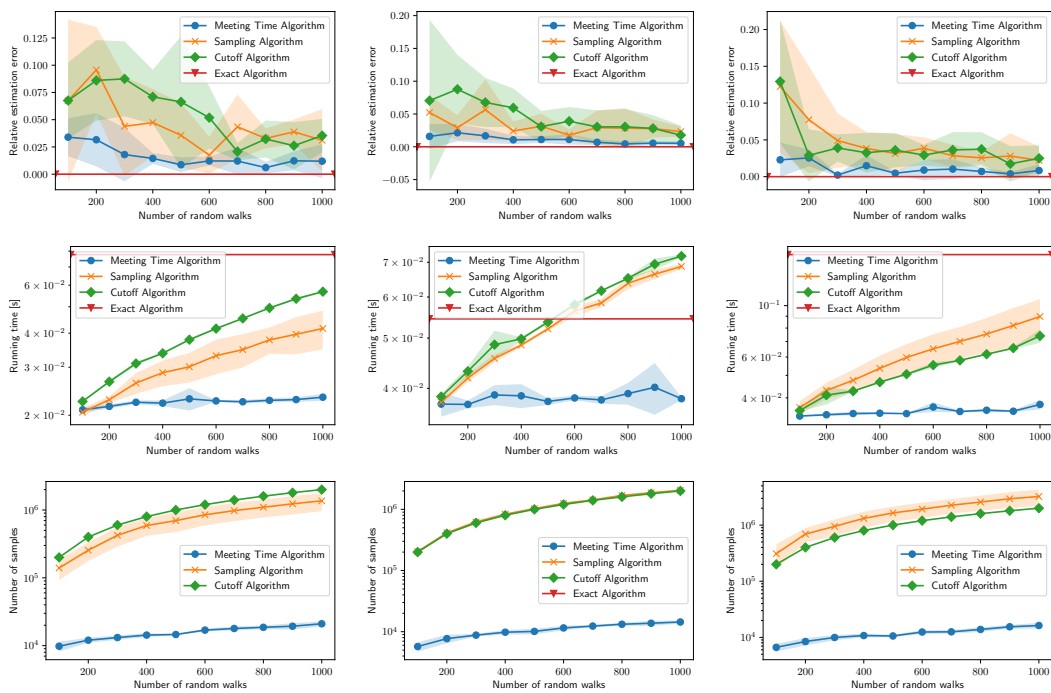

Figure 6: Ablation study for the number of random walks in synthetic datasets.

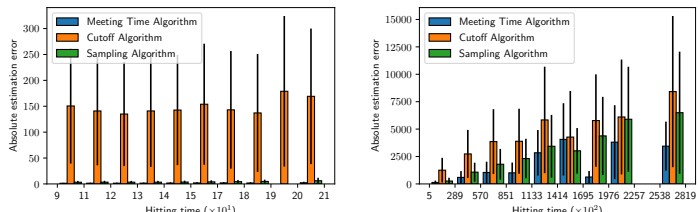

Figure 7: Absolute estimation error for uniformly sampled pairs on the Football (left) and Facebook (right) networks. The estimation errors are grouped by the true hitting time.

