# OpenReview forum: "Estimating Hitting Times Locally at Scale"
_NeurIPS.cc/2025/Conference — NeurIPS 2025 poster_

### Official Review · Reviewer_t7iu · 2025-06-28

**Clarity:** 2
**Significance:** 4
**Originality:** 4
**Rating:** 5
**Confidence:** 2

**Summary:**

This paper considers the problem of estimating hitting times in a simple random walk on a graph. The paper puts forth three approaches. The first (Algorithm 1) estimates hitting times through meeting times. The second (Algorithm 3) is based on a truncation of a hitting time summation representation, given in Lemma 4.1. The third approach simply samples long random walks and returns the average hitting time. Experiments on real and synthetic data are included.

**Questions:**

I did not follow the failure probability bound at the bottom of page 3. There is an intersection over \sigma, but in the text above, \sigma is defined to be a particular permutation.

Please address the point about Line 126.

Why does the Hoeffding bound apply in the line below 128? You are essentially conditioning on some events involving the A_i, B_i random variables, so you don’t have independence.

**Ethical Concerns:**

["NO or VERY MINOR ethics concerns only"]

**Final Justification:**

The authors addressed my concerns, so I raised my score.

**Limitations:**

yes

**Quality:**

4

**Strengths And Weaknesses:**

This paper provides two new approaches for hitting times estimation, given in Algorithms 1 and 3. There were some issues with clarity. In particular, I felt that Lemma 4.1 should be moved earlier, so that Lemma 3.3. would not need a forward reference.

My biggest confusion is in Lemma 3.4. Why is there a single meeting time T? I thought there were multiple meeting times? As a result I am not understanding the definitions of A_i and B_i, and therefore I am not seeing how the display below Line 126 is the output of Algorithm 1. Furthermore, the stopping condition of Algorithm 1 refers to $\chi$, but I don’t see $\chi$ defined.

Also, Lemma 3.3 refers to W, but W is not defined until Lemma 4.1.

There were a few cases where notation was not defined:
- Line 84: $\chi_{uv}$
- The set $\chi$ is not defined in Algorithm 1, so I cannot interpret the failure condition.

---

> ### Author Rebuttal · Authors · 2025-07-29
>
> We thank the reviewer for their thorough comments and positive feedback on the quality and significance of our work. Below, we address the main concerns raised, which primarily relate to clarity. We believe these issues can be effectively resolved in a revision and appreciate the reviewer’s helpful suggestions for improvement.
>
> **On Lemma 3.4** :
> * *“Why is there a single meeting time T?”*: In the definition of $A_i$ and $B_i$, we assume that the $i$-th random walk from $u$ meets and eliminates the $i$-th walk from $v$, which is without loss of generality since the walk label indices are arbitrarily chosen. We will denote the meeting time of each pair as $T_i$ to make clear that it is specific to each matched pair. The variables $A_i$ and $B_i$ count how many times the respective walks hit state $v$ before meeting. Line 126 in Algorithm 1 reflects this by incrementing or decrementing the hitting time estimate based on which walk visits $v$. We will update our write-up to make this important point clearer.
> * *“Why does the Hoeffding bound apply in the line below 128?”*: One of the ideas in our analysis is that the overall estimate can be written as a sum over $A_i - B_i$. As Lemma 3.4 conditions on successful meeting (i.e., within $t_{\max}$ steps), the random variables $|A_i - B_i|$ are bounded by $t_{\max}$ with (conditional) probability $1$. Combined with the fact that $|A_i - B_i|$ for all $i$ are independent, this justifies our application of Hoeffding’s inequality. We will make this clearer in the revision.
>
> **On Lemma 3.2 and the definition of $\sigma$**: The confusion likely stems from the wording. We take a conjunction over all possible permutations, and denote one such permutation by $\sigma$. We will clarify this in the proof to avoid ambiguity.
>
> **Other typos and clarifications:**
> * We will fix the typo in the stopping condition of Algorithm 1, replacing $\mathcal{X}$ with either $I$ or $J$, as appropriate.
> * We will move Lemma 4.1 between Lemmas 3.2 and 3.3 to avoid the forward reference and ensure that $W$ is properly defined.
> * Additional minor typos noted by the reviewer will also be addressed.
>
> We appreciate the reviewer’s close reading and helpful suggestions, which have improved the clarity and correctness of our presentation.

---

> > ### Comment · Reviewer_t7iu · 2025-08-04
> >
> > Thank you for addressing my concerns!

---

### Official Review · Reviewer_Sw5f · 2025-07-01

**Clarity:** 4
**Significance:** 4
**Originality:** 4
**Rating:** 6
**Confidence:** 3

**Summary:**

The work studies graphs as markov chains and formulate several methods to estimate hitting times using local state management of the graph random walks. Assuming steady state achievable markov chains on these graphs, they first relate chain mixing times to expected meeting times between 2 nodes/states on the graph. This helps them device parameters to decide on the estimated duration until which they run their algorithm (algorithm 1) and estimate hitting times based on several random walks they initiated from both the source and the target nodes. They also extend existing results on hitting times based on spectral values and tweak it into a local algorithm that runs on estimated local mixing time approximations. For all of these methods, authors provide algorithms, probabilistic error bounds, sample complexity bounds and run-time complexities. The focus was largely on the accuracy of the algorithm in achieving the true value in using fewer instances of random walks initiated by their algorithms. This was then backed by experiments on graphs where ground truth values of hitting times could be calculated.

**Questions:**

1. Algorithm 1 line 8, definition of $y_w$ should use $Y$ family of random walks to make sense.
2. Algorithm 1 line 14 should be some another variable, maybe $I$ or $J$ instead of $\chi$, right?
3. Moving from last expression in line 117 to 118 first expression, how did the $\pi_G(v)$ transform to $\pi_G(u)$?

**Ethical Concerns:**

["NO or VERY MINOR ethics concerns only"]

**Final Justification:**

The authors have addressed my concerns and subsequent clarification, actions from their side, and overall strengths and limitations of the paper has been discussed and agreed upon. As acknowledged before, I'll keep my score.

**Limitations:**

1. Extensions like algorithm 3 also face prohibitively large values of $r$, random walks with growth factors $\frac{m^2}{\epsilon^2}$. This is a crucial issue especially when one of the core assumptions is that the graph is connected and likely cannot be sparse!
2. While identified by authors themselves in their diligent analysis, expecting $t_{\text{mix}}$ as a one-shot input usually means running further sampling and approximation to estimate that first!

**Paper Formatting Concerns:**

Nothing major

**Quality:**

4

**Strengths And Weaknesses:**

Strength: 1. Rigorous theoretical analysis of several aspects of the proposed algorithm along with practical proof of concept.
2. At first glance step line 9 of Algorithm 1 looks prohibitively expensive, especially for large graphs, but the running time grows at most $\log^3n$, which is very promising for practice!

Weakness: 1. While they prove tight bounds on the samples required for the algorithms, $l$ that is the number of processes that they start random walk for, are practically large for large graphs. While authors mention that one of the core motivations was to avoid using global algorithms that prohibit working with large graphs, the design choice of $l$ (which comes with excellent guarantees as well), is proportional to squared total edges in the graph as well as inverse square of the accuracy error. While the local approach allows them to not store all these walks, even storing the current state of these many walks incurs huge memory as well, albeit the memory requirements decline as the algo does away with intersected random walks.

---

> ### Author Rebuttal · Authors · 2025-07-29
>
> We thank the reviewer for their positive feedback and helpful comments.
>
> The reviewer correctly noted that the pseudocode appears, at first glance, to loop over all nodes. While this presentation was chosen for simplicity, we agree that it is misleading in retrospect. The algorithm actually only iterates over a significantly smaller subset of nodes. We will revise the pseudocode and accompanying explanation to make this clear.
>
> Regarding the worst-case performance of our algorithms: we agree with the reviewer that the mixing time plays a central role in the time complexity and can indeed become a bottleneck in some regimes. Obtaining tight bounds for estimating hitting times remains an interesting open problem. Designing average-case algorithms for such problems is yet another such promising direction for future research.
>
> Finally, we will address the specific corrections the reviewer kindly pointed out:
> * In Algorithm 1, we will replace the incorrect terminating condition symbol $\chi$ with $I$ or $J$, as suggested.
> * We will fix the typo on Line 8 of Algorithm 1.
> * In Lines 117–118, we will correct the expression to $\pi(v)$.
>
> We thank the reviewer again for their close reading and valuable suggestions.

---

> > ### Comment · Reviewer_Sw5f · 2025-08-04
> >
> > I thank the authors for their response. I have also taken into account the comments from other reviewers. As discussed in this particular review above, both sides agree on the raised concerns, clearly recognize limitations, and minor corrections to improve clarify. Therefore, I’ll maintain my score.

---

### Official Review · Reviewer_fb21 · 2025-07-02

**Clarity:** 3
**Significance:** 3
**Originality:** 3
**Rating:** 5
**Confidence:** 2

**Summary:**

This paper introduces local algorithms for estimating the hitting time $H_G(u,v)$ between two vertices.  The central idea is simple: consider the meeting time of two independent random walks originating at different vertices, $u$ and $v$.  Because the infinite series that defines a hitting time collapses once the walks intersect, the resulting estimator is unbiased, comes with explicit additive‑error guarantees, and provably uses nearly the minimum possible number of samples.

Beyond this meeting‑time approach, the authors adapt the spectral‑cutoff method of Peng et al. to the asymmetric case.  They provide matching lower bounds that justify why straightforward walk sampling cannot achieve the same accuracy, and they reinterpret hitting‑time estimation as a property testing problem, swapping global mixing‑time parameters for quantities that can be measured locally.

**Questions:**

Please refer to the weakness section

**Ethical Concerns:**

["NO or VERY MINOR ethics concerns only"]

**Final Justification:**

The paper presents a thorough theoretical examination of various aspects of the proposed algorithm, supported by a practical proof of concept. The exploration of local or sublinear graph algorithms is an exciting and valuable research direction that merits greater attention. Having read the rebuttal, I have decided to maintain my score.

**Limitations:**

yes

**Paper Formatting Concerns:**

No major formatting concerns

**Quality:**

3

**Strengths And Weaknesses:**

Strengths:

The results are backed up with strong theory: Provides additive‑error guarantees, a near‑optimal lower bound, an asymmetric spectral‑cutoff analysis, and a reduction to a property testing problem.

Weakness:

1) It would strengthen the paper to include experiments on graphs of at least 100k nodes (and ideally larger) to demonstrate scalability.

2) Adding empirical comparisons against common heuristics, such as personalized or approximate PageRank, would help illustrate the method’s practical relevance.

---

> ### Author Rebuttal · Authors · 2025-07-29
>
> We thank the reviewer for their time and thoughtful comments.
> * We agree that experiments on even larger graphs would help further demonstrate the scalability of our approach. As per the reviewer’s suggestion, we will include experiments on even larger graphs in our revision. We expect the results to mimic the trend of our current experimental setup.
> * We also appreciate the reviewer’s suggestion that our work’s practical relevance could be better contextualized through comparison with approaches for related measures such as Personalized PageRank (PPR) [1]. We will add a discussion of this point in the revised version. This is a very promising direction for future work as well!
>
> We thank the reviewer again for raising these important points. In an extended version of the paper, we plan to include experiments on larger datasets and explore using PPR as a proxy baseline for our algorithms.
>
> [1] Chung, Fan, and Olivia Simpson. "Computing heat kernel pagerank and a local clustering algorithm." European Journal of Combinatorics 68 (2018): 96-119.

---

> > ### Comment · Reviewer_fb21 · 2025-08-02
> >
> > Thank you for your rebuttal. I will keep my score.

---

### Official Review · Reviewer_BMmT · 2025-07-03

**Clarity:** 3
**Significance:** 4
**Originality:** 3
**Rating:** 5
**Confidence:** 4

**Summary:**

This paper is on estimating the hitting time in an undirected graph. Classic algorithms that compute the hitting time run in O(n^3) time. This paper gives a random walk based algorithm that estimates the hitting time. The core idea is to use (what I would call) a "bidirectional estimator". To estimate the hitting time from u to v, one runs random walks from both u and v, look at the walks that "collide", and scale that to get an estimate. The difficult part is usually in bounding the concentration and proving that the estimate converges with a few samples.

Essentially, when the mixing time is small (much smaller than the hitting time), then the bidirectional estimates converge rapidly. This can be shown using matrix Chernoff bounds.

The running time of this estimator is expressed in terms of the mixing time and the stationary distribution. In the worst case, the running time significantly worse than O(n^3). If the graph is an expander (mixing time = O(\log n)), then the running time is better than the worst cast (though still worse than n^3?). If the degree distribution is skewed, then I think there is a significant advantage.

The result also gives empirical experiments showing that the algorithm works well in practice, which I think is nice.

**Questions:**

1. As said earlier: please give an explanation of when the algorithm is better than n^3, and the impact of skewed pi(v) values.
2. There is a lot of literature on similar estimators used for PageRank, PersonalizedPageRank, and estimating the stationary distribution. I would recommend citing some of these papers, and the papers that they cite.
 - FAST-PPR: Scaling Personalized PageRank Estimation for Large Graphs, Peter Lofgren, Siddhartha Banerjee, Ashish Goel, and C. Seshadhri, KDD 2014.
 - Personalized PageRank Estimation and Search: A Bidirectional Approach. Peter Lofgren, Siddhartha Banerjee, Ashish Goel. WSDM 2016
 - Fast Bidirectional Probability Estimation in Markov Models. Siddhartha Banerjee, Peter Lofgren. NIPS 2015
 - Estimating Single-Node PageRank in O ̃ (min{dt,√m}) Time. Hanzhi Wang and Zhewei Wei, VLDB 2023
 - Efficient Algorithms for Personalized PageRank Computation: A Survey. Mingji Yang, Hanzhi Wang, Zhewei Wei, Sibo Wang, and Ji-Rong Wen. TKDE 2024
 - Revisiting Local PageRank Estimation on Undirected Graphs: Simple and Optimal. Hanzhi Wang. KDD 2024

**Ethical Concerns:**

["NO or VERY MINOR ethics concerns only"]

**Limitations:**

Yes

**Quality:**

3

**Strengths And Weaknesses:**

Getting local/sublinear graph algorithms is a fantastic topic and needs more focus. I think the hitting time is an excellent testbed for such algorithm. The bidirectional estimator has shown much promise in theory and practice, and it's great to see more applications of this technique.

The major weakness is a lack of an explanation of when this algorithm works. Theorem 3.1 and Corollary 3.1 should be accompanied with worst-case running times (in terms of n), an explanation of the rapidly mixing case, and possible of the skewed degree distribution case (so that \pi(v) can be much larger than 1/n). I believe that the latter is the reason why the algorithm works well in practice.

---

> ### Author Rebuttal · Authors · 2025-07-29
>
> We thank the reviewer for their thoughtful and constructive feedback.
> * We agree that a more detailed discussion of worst-case running times for our algorithms under various parameter regimes would improve the exposition. In Theorem 3.1, the mixing time $t_{\text{mix}}$ can reach $n^3$ in the worst case, while $\pi(v) \geq 1/n^2$ and $|\pi_G|_2 \geq \sqrt{1 / n}$. However, these worst-case bounds are not independent of each other, making a direct maximization of the expression nontrivial. For this reason, we chose to express the running time in terms of the mixing time and stationary distribution parameters. We also considered upper bounding the mixing time using the conductance $\Phi$ [1] and the maximum degree $\Delta$. We will include a discussion of these points in our revision.
> * We agree that graphs with skewed degree distributions are an important regime to study, often allowing our algorithms to perform much more efficiently compared to the worst case. Obtaining tight bounds for such graphs is a promising direction for future work.
> * Finally, we appreciate the reviewer’s suggestions regarding related literature on Personalized PageRank. We agree that such related measures are a natural point of comparison for our algorithms, and we will incorporate a more thorough discussion of this connection in our revision.
>
> [1] Lovász, László, and Ravi Kannan. "Faster mixing via average conductance." Proceedings of the thirty-first annual ACM symposium on Theory of computing. 1999.

---

### Decision · Program_Chairs · 2025-09-17

**Decision:**

Accept (poster)

**Comment:**

One of the key quantities that can be attached to a random walk over a graph $G$ is the (expected) hitting time of a vertex $v$ when the random walk starts from vertex $u$. This paper provides a "local" method to compute such quantities. To be more specific: by performing lots of independent random walks on the graph, and keeping track of which pairs of random walks have met (and occasionally "killing" some of the random walks), one gets a good additive estimate of the hitting time. The main results still allow for a large time complexity -- i.e. a large total number of RW steps -- but two points must be added here. One is that, on skewed degree distributions, the theoretical bounds actually perform better. The other is about space complexity: the fact that the method is "local" means that one never needs to load the full graph into memory. The paper also contains a number of relatively small-scale experiments (restricted to graphs with 10^4 nodes) that suggest its method (or rather, methods) perform well.

The strengths of the paper are numerous. As noted above, there are clear advantages to using "local" algorithms. The present paper builds on related mathematical works on random walks to obtain interesting applicable methods. This theoretical analysis is interesting, although "wasteful" in that it is very unlikely that e.g. A_i-B_i ever reaches t_{\max} (to mention one very technical point). In fact, I think that much of the success in the experiments can be credited to this feature. The notion of "local mixing time" introduced in the analysis of the second method would be an interesting object for further study.

As for weaknesses, some typos and slightly imprecise steps in the analysis were identified (see comments by Sw5f and t7iU). It seems to me that all of these can be easily fixed. In the present form, the paper also lacks more large scale experiments (one reviewer suggested graphs with 10^5 nodes). The authors have also suggested that they will add such experiments, but it remains to be seen what there will look like. Another weakness is that its theme is not quite central to NeurIPS. About this, however, I would say that the paper is at the very least related to the analysis of large networks, which is an important direction in statistics and ML. Therefore, my evaluation is that these weaknesses are not serious.

To sum up, I agree with the reviewers' positive assessment of this submission. As noted above, it makes interesting contributions in terms of theory, and the experimental evidence, while not quite conclusive (due to the small network sizes), looks promising. (The discussion was uneventful, which I see as a sign that the paper is well written!)